# An Analysis of Model-Based Reinforcement Learning From Abstracted Observations

**Rolf A. N. Starre**  *r.a.n.starre@tudelft.nl*
*Delft University of Technology*

**Marco Loog**  *marco.loog@ru.nl*
*Radboud University*

**Elena Congeduti**  *e.congeduti@tudelft.nl*
*Delft University of Technology*

**Frans A. Oliehoek**  *f.a.oliehoek@tudelft.nl*
*Delft University of Technology*

**Reviewed on OpenReview:** *https://openreview.net/forum?id=YQWOzzSMPp*

## Abstract

Many methods for Model-based Reinforcement learning (MBRL) in Markov decision processes (MDPs) provide guarantees for both the accuracy of the model they can deliver and the learning efficiency. At the same time, state abstraction techniques allow for a reduction of the size of an MDP while maintaining a bounded loss with respect to the original problem. Therefore, it may come as a surprise that no such guarantees are available when combining both techniques, i.e., where MBRL merely observes abstract states. Our theoretical analysis shows that abstraction can introduce a dependence between samples collected online (e.g., in the real world). That means that, without taking this dependence into account, results for MBRL do not directly extend to this setting. Our result shows that we can use concentration inequalities for martingales to overcome this problem. This result makes it possible to extend the guarantees of existing MBRL algorithms to the setting with abstraction. We illustrate this by combining R-MAX, a prototypical MBRL algorithm, with abstraction, thus producing the first performance guarantees for model-based 'RL from Abstracted Observations': model-based reinforcement learning with an abstract model.

## 1 Introduction

Tabular Model-based Reinforcement Learning (MBRL) methods provide guarantees that show they can learn efficiently in Markov decision processs (MDPs) (Brafman & Tennenholtz, 2002; Strehl & Littman, 2008; Jaksch et al., 2010; Fruit et al., 2018; Talebi & Maillard, 2018; Zhang & Ji, 2019; Bourel et al., 2020). They do this by finding solutions to a fundamental problem for Reinforcement Learning (RL), the exploration-exploitation dilemma: when to take actions to obtain more information (explore) and when to take actions that maximize reward based on the current knowledge (exploit). However, MDPs can be huge, which can be problematic for tabular methods. One way to deal with large problems is by using abstractions, such as the mainstream state abstractions (Li, 2009; Abel et al., 2016). State abstractions reduce the size of the problem by aggregating together states according to different criteria, depending on the specific type of abstraction. We can view state abstraction as a special case of function approximation, where every state maps to its abstract state (Mahadevan, 2010), and we can roughly divide them into *exact* and *approximate* abstractions (Li, 2009; Abel et al., 2016).

Approximate abstractions relax the criteria of exact abstractions, and therefore allow for a larger reduction in the state space. Typically, this approximation leads to a trade-off between performance and the amount of required data (Paduraru et al., 2008; Jiang et al., 2015). In this paper, we will assume the use of abstraction as a given, e.g., because the complete state space is too large to deal with. Nevertheless, we explore the trade-off in Section 4, where we compare the performance of the prototypical R-MAX algorithm (Brafman & Tennenholtz, 2002) with and without abstraction.

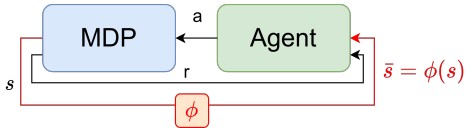

Figure 1: RL from Abstracted Observations, the agent receives the abstract state $\bar{s} = \phi(s)$ as an observation instead of the state $s$. Image based on Abel et al. (2018).

In our setting, the agent acts in an MDP that returns states $s$, but instead of observing the true state $s$, the agent only observes abstract states $\phi(s)$ (see Figure 1). This setting, which has been considered before (Ortner et al., 2014a; Abel et al., 2018),[1] is what we call *RL from Abstracted Observations (RLAO)*. Surprisingly, there are relatively few results for RLAO, even though many results for the planning setting are available (Li et al., 2006; Abel et al., 2016). The main difference between these two settings is that in planning with abstraction the resulting problem can still be considered an MDP, but in RLAO, while the underlying problem is still an MDP, the observed problem is not.

The observation that the observed problem is not an MDP can be understood when we realize that RLAO corresponds to RL in a Partially Observable MDP (POMDP) (Kaelbling et al., 1998), as previously described (Bai et al., 2016). Specifically, the abstraction function serves as an observation function. Rather than observing its true state $s$, the agent observes the abstract state $\phi(s)$ and its policy chooses an action based on this abstract state. It is well known that policies for POMDPs that only base their action on the last observation can be arbitrarily bad (Singh et al., 1994). Fortunately, there is also good news, as this worst-case does not apply when $\phi$ is an *exact model similarity abstraction* [2] (Li, 2009), because the resulting problem can be considered an MDP; this abstraction maps states to the same abstract state only when their reward and transition functions in the abstract space are the same (Li et al., 2006). We focus on the related *approximate model similarity abstraction* (Abel et al., 2016), which maps states to the same abstract state only when their reward and transition functions in the abstract space are close. Intuitively, because of its connection to the exact model similarity, one could expect that for this abstraction the worst-case also does not apply. However, as we discuss in detail in Section 2.2, MBRL methods typically use results that rely on the assumption of independent and identically distributed (i.i.d.) samples to prove efficient learning (Strehl & Littman, 2008; Jaksch et al., 2010; Fruit et al., 2018; Bourel et al., 2020). This is not appropriate in RLAO: with abstraction, the transitions between abstract states need not be Markov, and the samples may depend on the history.

We analyze collecting samples in RLAO and prove that, with abstraction, samples are not guaranteed to be independent. This means that *most guarantees of existing MBRL methods do not hold in the RLAO setting.* [3] The primary technical result in this work shows that we can still learn an accurate model in RLAO by replacing concentration inequalities that rely on independent samples with a well-known concentration inequality for martingales (Azuma, 1967). This result allows us to extend the guarantees of MBRL methods to RLAO. We illustrate such an extension for the prototypical R-MAX algorithm (Brafman & Tennenholtz, 2002), thus producing the first performance guarantees for model-based methods in RLAO. These results are important for the often adopted state abstraction framework, as they allow us to conclude under what cases performance guarantees in MBRL can be transferred to settings with state abstraction.

---

[1]We refer to Section 5 for a comparison with the related work.

[2]Also known as stochastic bisimulation (Givan et al., 2003).

[3]Of course, certain guarantees on the combination of abstraction and RL are known. However, in most related work in abstraction settings (e.g., abstraction selection), the complication of samples not being independent does not occur due to particular assumptions (Paduraru et al., 2008; Hallak et al., 2013; Maillard et al., 2013; Majeed & Hutter, 2018; Ortner et al., 2019; Du et al., 2019). Section 5 gives details for individual papers.

## 2 Background

Section 3 will cover the combination of MBRL and abstraction in MDPs, in this section we introduce the required background.

### 2.1 Model-Based RL

As is typical for RL problems, we assume the environment the agent is acting in can be represented by an infinite horizon MDP $M \triangleq \langle S, A, T, R, \gamma \rangle$ (Puterman, 2014). Here $S$ is a finite set of states $s \in S$, $A$ a finite set of actions $a \in A$, $T$ a transition function $T(s'|s,a) = \Pr(s'|s,a)$, $R$ a reward function $R(s,a)$ which gives the reward received when the agent executes action $a$ in state $s$, and $\gamma$ is a discount factor with $0 \le \gamma \le 1$ that determines the importance of future rewards. We use $R_{\max}$ to denote the maximum reward the agent can obtain in one step. The agent's goal is to find an optimal policy $\pi^* : S \to A$, i.e., a policy that maximizes the expectation of the cumulative reward in the MDP. $V^\pi(s)$ denotes the expected value of the cumulative reward under policy $\pi$ starting from state $s$. Similarly, $Q^\pi(s,a)$ denotes the expected value of the cumulative reward when first taking action $a$ from state $s$ and then following policy $\pi$ afterward.

MBRL methods learn a model from the experience that the agent gains by taking actions and observing the rewards it gets and the states it reaches. For a fixed state-action pair $(s,a)$, we let $\tau_1, \tau_2, \cdots, \tau_{N(s,a)}$ be the first $N(s,a)$ time steps at which the agent took action $a$ in state $s$. The first $N(s,a)$ states $s'$ that the agent reached after taking action $a$ in state $s$ are stored as the sequence $Y_{s,a} \triangleq (s'^{(\tau_1+1)}, s'^{(\tau_2+1)}, \cdots, s'^{(\tau_{N(s,a)}+1)})$. We use $Y$ to refer to the collection of all $Y_{s,a}$. Typically, in MBRL, the obtained experience is used to construct the empirical model $T_Y$ (Brafman & Tennenholtz, 2002; Strehl & Littman, 2008; Jaksch et al., 2010; Fruit et al., 2018; Talebi & Maillard, 2018; Zhang & Ji, 2019; Bourel et al., 2020). This model is constructed simply by counting how often the agent reached a particular next state $s'$ and normalizing the obtained quantity by the total count:

$$\forall s' \in S : T_Y(s'|s,a) \triangleq \frac{1}{N(s,a)} \sum_{i=1}^{N(s,a)} \mathbb{1}\{Y_{s,a}^{(\tau_i+1)} = s'\}. \tag{1}$$

Here $\mathbb{1}\{\cdot\}$ denotes the indicator function of the specified event, i.e., $\mathbb{1}\{Y_{s,a}^{(\tau_i+1)} = s'\}$ is 1 if $Y_{s,a}^{(\tau_i+1)} = s'$ and 0 otherwise.

### 2.2 Guarantees for MBRL

The quality of the empirical model $T_Y$ is crucial for performance guarantees, irrespective of the form of the guarantee, e.g., PAC-MDP (Strehl & Littman, 2008) or regret (Jaksch et al., 2010). The quality of the empirical model is high when the distance between $T_Y(\cdot|s,a)$ and the ground truth $T(\cdot|s,a)$ is small. We can, for instance, measure this distance with the $L_1$ norm, defined as follows:

$$||T_Y(\cdot|s,a) - T(\cdot|s,a)||_1 \triangleq \sum_{s' \in S} |T_Y(s'|s,a) - T(s'|s,a)|. \tag{2}$$

Concentration inequalities are often used to guarantee that, with enough samples, this distance will be small, e.g.:

**Lemma 1** ($L_1$ inequality (Weissman et al., 2003)). *Let $Y_{s,a} = Y_{s,a}^{(1)}, Y_{s,a}^{(2)}, \cdots, Y_{s,a}^{(N(s,a))}$ be i.i.d. random variables distributed according to $T(\cdot|s,a)$. Then, for all $\epsilon > 0$,*

$$\Pr(||T_Y(\cdot|s,a) - T(\cdot|s,a)||_1 \ge \epsilon) \le (2^{|S|} - 2)e^{-\frac{1}{2}N(s,a)\epsilon^2}. \tag{3}$$

These inequalities typically make use of the fact that samples are i.i.d. It is not necessarily evident that these bounds can be applied without problem. Let us explore the transitions from a particular state, say state 42, in a Markov chain (we can ignore actions for this argument). Let $k$ and $l$ denote the time steps of

two different visits to state 42. Without abstraction, the conditional distributions from which next states are sampled are identical. So the question now is if these are independent. That is, is it the case that:

$$P(S_{k+1}, S_{l+1}|S_k = 42, S_l = 42) = P(S_{k+1}|S_k = 42) * P(S_{l+1}|S_l = 42)? \tag{4}$$

We have that

$$P(S_{k+1}, S_{l+1}|S_k = 42, S_l = 42) = P(S_{k+1}|S_k = 42, S_l = 42)P(S_{l+1}|S_k = 42, S_k + 1, S_l = 42) \tag{5}$$
$$= P(S_{k+1}|S_k = 42, S_l = 42)P(S_{l+1}|S_l = 42) \text{ (due to the Markov property)} \tag{6}$$

So the question is if $P(S_{k+1}|S_k = 42, S_l = 42) = P(S_{k+1}|S_k = 42)$? In general, this is not the case, since the information that $S_l = 42$ gives information about what $S_{k+1}$ was.

However, as shown for instance by Strehl & Littman (2008), concentration inequalities for i.i.d. samples, such as Hoeffding's Inequality, can still be used as an upper bound in this case, because of the Markov property and the identical distributions of the samples. In this way, MBRL can upper bound the probability that the empirical model $T_Y(\cdot|s, a)$ will be far away ($\geq \epsilon$) from the actual model $T(\cdot|s, a)$. When the empirical model is accurate, a policy based on this model leads to near-optimal performance in the MDP $M$ (Brafman & Tennenholtz, 2002; Strehl & Littman, 2008; Jaksch et al., 2010; Bourel et al., 2020).

## 2.3 State Abstraction for Known Models

We can formulate state abstraction as a mapping from states to abstract states (Li et al., 2006). This mapping is done with an abstraction function $\phi$, a surjective function that maps from states $s \in S$ to abstract states $\bar{s} \in \bar{S}$: $\phi(s) : S \rightarrow \bar{S}$. We use the $\bar{\phantom{x}}$ notation to refer to the abstract space and define $\bar{S}$ as $\bar{S} = \{\phi(s)|s \in S\}$. We slightly overload the definition of $\bar{s}$ to be able to write $s \in \bar{s}$. In this case, $\bar{s}$ is the set of states that map to $\bar{s}$, i.e., $\bar{s} = \{s \in S \mid \phi(s) = \bar{s}\}$. This form of state abstraction is general, and clusters states with different dynamics into abstract states. We assume that the state abstraction deterministically maps states to an abstract state. Since each state maps to precisely one abstract state and multiple states can map to the same abstract state, the abstract state space is typically (much) smaller than the original state space, $|\bar{S}| \leq |S|$.

We focus on a type of abstraction *approximate model similarity abstraction* (Abel et al., 2016), also known as approximate stochastic bisimulation (Dean et al., 1997; Givan et al., 2003). In this abstraction, two states can map to the same abstract state only if their behavior is similar in the abstract space, i.e., when the reward function and the transitions to abstract states are close. We can determine the transition probability to an abstract state $T(\bar{s}'|s, a)$ as:

$$T(\bar{s}'|s, a) = \sum_{s' \in \bar{s}'} T(s'|s, a). \tag{7}$$

Then, we can use equation 7 to define approximate model similarity abstraction:

**Definition 1.** *An approximate model similarity abstraction, $\phi_{model, \eta_R, \eta_T}$, for fixed $\eta_R, \eta_T$, satisfies*

$$\phi_{model, \eta}(s_1) = \phi_{model, \eta}(s_2) \implies \forall a \in A : |R(s_1, a) - R(s_2, a)| \leq \eta_R,$$
$$\forall \bar{s}' \in \bar{S}, a \in A : |T(\bar{s}'|s_1, a) - T(\bar{s}'|s_2, a)| \leq \eta_T. \tag{8}$$

From now on, we will refer to $\phi_{model, \eta_R, \eta_T}$ as $\phi$. We note that this abstraction is still quite generic. It can cluster together states that have different transition and reward functions.

## 2.4 Planning With Abstract MDPs

In the planning setting, where the model is known a priori, we can use the abstraction function $\phi$ to construct an abstract MDP. An abstract MDP can be helpful because it is smaller, making it easier to find a solution,

and a solution for the abstract MDP can work well in the original MDP (Li et al., 2006; Abel et al., 2016). We construct an abstract MDP $\bar{M}_\omega$ from the model of an MDP $M$, an abstraction function $\phi$, and an action-specific weighting function $\omega$. [4] The weighting function $\omega$ gives a weight to every state-action pair: $\forall s \in S, \ a \in A : 0 \leq \omega(s, a) \leq 1$. The weights of the state-action pairs associated with an abstract state $\bar{s}$ sum up to 1: $\sum_{s' \in \phi(s)} \omega(s', a) = 1$. We can use the weighting function to create an abstract transition and reward function, which are weighted averages of the original transition and reward functions. In this way, from $M$, $\phi$, and any $\omega$, we can *construct* an abstract MDP $\bar{M}_\omega$:

**Definition 2** (Abstract MDP). *Given an MDP $M$, $\phi$, and $\omega$, an abstract MDP $\bar{M}_\omega = \langle \bar{S}, A, \bar{T}_\omega, \bar{R}_\omega \rangle$ is constructed as: $\bar{S} = \{\phi(s) \mid s \in S\}, A = A,$*

$$\forall \bar{s} \in \bar{S}, \ a \in A : \bar{R}_\omega(\bar{s}, a) = \sum_{s \in \bar{s}} \omega(s, a) R(s, a), \tag{9}$$

$$\forall \bar{s}, \bar{s}' \in \bar{S}, \ a \in A : \bar{T}_\omega(\bar{s}'|\bar{s}, a) = \sum_{s \in \bar{s}} \sum_{s' \in \bar{s}'} \omega(s, a) T(s'|s, a). \tag{10}$$

Note that the abstract MDP $\bar{M}_\omega$ itself is an MDP. So we can use planning methods for MDPs to find an optimal policy $\bar{\pi}^*$ for $\bar{M}_\omega$. A desirable property of the approximate model similarity abstraction is that we can upper bound the difference between the optimal value $V^*$ in $M$ and the value $V^{\bar{\pi}^*}$ obtained when following the policy $\bar{\pi}^*$ in $M$. These bounds exists in different forms (Dearden & Boutilier, 1997; Abel et al., 2016; Taïga et al., 2018). For completeness, we give these bounds for both the undiscounted finite horizon and the discounted infinite horizon:

**Theorem 1.** *Let $M = \langle S, A, T, R \rangle$ be an MDP and $\bar{M} = \langle \bar{S}, A, \bar{T}, \bar{R} \rangle$ an abstract MDP, for some defined abstract transitions and rewards. We assume that*

$$\forall \bar{s}, \bar{s}' \in \bar{S}, \ s \in \bar{s}, \ a \in A : |\bar{T}(\bar{s}'|\bar{s}, a) - \Pr(\bar{s}'|s, a)| \leq \eta_T \tag{11}$$

$$and \ |\bar{R}(\bar{s}, a) - R(s, a)| \leq \eta_R. \tag{12}$$

*Then, for a finite horizon problem with horizon $h$ we have:*

$$V^*(s) - V^{\bar{\pi}^*}(s) \leq 2h\eta_R + (h+1)h\eta_T|\bar{S}|R_{max}. \tag{13}$$

*And for a discounted infinite horizon problem with discount $\gamma$ we have:*

$$V^*(s) - V^{\bar{\pi}^*}(s) \leq \frac{2\eta_R}{1-\gamma} + \frac{2\gamma\eta_T|\bar{S}|R_{max}}{(1-\gamma)^2}. \tag{14}$$

The proof of Theorem 1 is in Appendix A.3. These bounds show that an optimal abstract policy $\bar{\pi}^*$ for $\bar{M}$ can also perform well in the original problem $M$ when the approximate errors $\eta_R$ and $\eta_T$ are small. They hold for any abstract MDP $\bar{M}$ created from an approximate model similarity abstraction $\phi$ and any valid weighting function $\omega$.

## 3 MBRL From Abstracted Observations

In RLAO, we have an abstraction function $\phi$ and instead of observing the true state $s$, the agent observes the abstract state $\phi(s)$. In contrast to the planning setting in Section 2.3, here we act in an MDP $M$ of which we do *not* know the transition and reward functions. As mentioned in the introduction, there are surprisingly few results for the RLAO setting (Section 5 discusses special cases people have considered). Specifically, results of MBRL from Abstracted Observations (MBRLAO) are lacking. Section 3.2 explains why this is by analyzing how abstraction leads to dependence between samples, which means that the methods for dealing with Markov transitions, as covered in Section 2.2, no longer suffice. Then, in Section 3.4, we show how concentration inequalities for martingales can be used to still learn an accurate model in RLAO. To illustrate how this result can be used to extend the results of MBRL methods to RLAO, we extend the results of the R-MAX algorithm (Brafman & Tennenholtz, 2002). R-MAX is a well-known and straightforward method that guarantees sample efficient learning.

---

[4] The action-specific weighting function is more general than the typically used weighting function, which is not action-specific and only depends on the state $s$ (Li et al., 2006). More formally, it is the case where $\forall a, a' \in A, \ s \in S : \omega(s, a) = \omega(s, a')$.

### 3.1 The General MBRL From Abstracted Observations Approach

In RLAO, the agent collects data for every abstract state-action pair $(\bar{s}, a)$, stored as sequences $\bar{Y}_{\bar{s},a}$:

$$\bar{Y}_{\bar{s},a} : \left\{ \bar{s}'^{(\tau_1+1)}, \bar{s}'^{(\tau_2+1)}, \cdots, \bar{s}'^{(\tau_{N(\bar{s},a)}+1)} \right\}. \tag{15}$$

Like in equation 1, we construct an empirical model $\bar{T}_Y$, now looking at the abstract next-states that the agent reached:

$$\bar{T}_Y(\bar{s}'|\bar{s}, a) \triangleq \frac{1}{N(\bar{s},a)} \sum_{i=1}^{N(\bar{s},a)} \mathbb{1}\{\bar{Y}_{\bar{s},a}^{(i)} = \bar{s}'\}. \tag{16}$$

Suppose we could guarantee that the empirical model $\bar{T}_Y$ was equal, or close, to the transition function $\bar{T}_\omega$ of an abstract MDP $\bar{M}_\omega$ constructed from the true MDP with $\phi$ and a valid $\omega$. In that case, we could bound the loss in performance due to applying the learned policy $\bar{\pi}^*$ to $M$ instead of applying the optimal policy $\pi^*$ (Abel et al., 2016; Taïga et al., 2018). Our main question is: do the finite-sample model learning guarantees of MBRL algorithms still hold in the RLAO setting?

### 3.2 Requirements for guarantees for MBRL From Abstracted Observations

In order to give guarantees, we need to show that the empirical model $\bar{T}_Y$ is close to the transition model of an abstract MDP $\bar{M}_\omega$. Before defining this transition model of $\bar{M}_\omega$, we examine the data collection. In the online data collection, the agent obtains a sample for $\bar{Y}_{\bar{s},a}$ when it is in a state $s \in \bar{s}$ and takes action $a$. Specifically, the agent obtains the $i$-th sample $\bar{Y}_{\bar{s},a}^{(i)} = \bar{s}'^{\tau_1+1}$ from state $X_{\bar{s},a}^{(i)} = s^{\tau_i} \in \bar{s}$:

$$\bar{Y}_{\bar{s},a}^{(i)} \sim T(\cdot|X_{\bar{s},a}^{(i)} = s^{\tau_i}, a). \tag{17}$$

Let $X_{\bar{s},a} = (X_{\bar{s},a}^{(i)})_{i=1}^{N(\bar{s},a)}$ denote the sequence of states $s \in \bar{s}$ from which the agent took action $a$. Each state $s$ gets a weight according to how often it appears in $X_{\bar{s},a}$, which we formalize with the weighting function $\omega_X$:

$$\forall s \in \bar{s}, a \in A : \omega_X(s,a) \triangleq \frac{1}{N(\bar{s},a)} \sum_{i=1}^{N(\bar{s},a)} \mathbb{1}\{X_{\bar{s},a}^{(i)} = s\}. \tag{18}$$

We use $\omega_X$ to define $\bar{T}_{\omega_X}$ analogous to equation 10:

$$\forall \bar{s}, \bar{s}' \in \bar{S}, a \in A : \bar{T}_{\omega_X}(\bar{s}'|\bar{s}, a) \triangleq \sum_{s \in \bar{s}} \omega_X(s,a) \sum_{s' \in \bar{s}'} T(s'|s, a). \tag{19}$$

To highlight the close connection between $\bar{T}_{\omega_X}$ and $\bar{T}_Y$ (build of samples from $T(\cdot|X_{\bar{s},a}^{(i)} = s^{\tau_i}, a)$), we give a second, but equivalent, [5] definition of $\bar{T}_{\omega_X}$:

$$\forall (\bar{s}, a), \bar{s}' : \bar{T}_{\omega_X}(\bar{s}'|\bar{s}, a) \triangleq \frac{1}{N(\bar{s},a)} \sum_{i=1}^{N(\bar{s},a)} T(\bar{s}'|X_{\bar{s},a}^{(i)}, a). \tag{20}$$

Note that $\omega_X$ and thus $\bar{T}_{\omega_X}$ are not fixed a priori. Instead, like $\bar{T}_Y$, they are empirical quantities that change at every time step and depend on the policy and the (stochastic) outcomes. Importantly, by its definition, $\omega_X$ is a valid $\omega$ at every timestep. It is not a problem that $\omega_X$ and $\bar{T}_{\omega_X}$ change over time, as long as the empirical model $\bar{T}_Y$ can be shown to be close to $\bar{T}_{\omega_X}$. For this, we want a concentration inequality to provide bounds on the deviation of the empirical model $\bar{T}_Y$ from $\bar{T}_{\omega_X}$; we refer to this inequality as the abstract L1 inequality, similar in form to equation 3:

$$P(|\bar{T}_Y(\cdot|\bar{s}, a) - \bar{T}_{\omega_X}(\cdot|\bar{s}, a)|_1 \geq \epsilon) \leq \delta, \tag{21}$$

where $\bar{T}_Y(\cdot|\bar{s}, a)$ is defined according to equation 16 and $\bar{T}_{\omega_X}$ according to equation 19.

---

[5]In the proof of Theorem 2 we show that these two definitions are equivalent.

### 3.3 Why the Previous Strategy Fails: Dependent Samples That Are Not Identically Distributed

Suppose we could directly obtain i.i.d. samples from $\bar{T}_{\omega_X}$ and base our empirical model $\bar{T}_Y$ on the obtained samples. In that case, we could show that the abstract L1 inequality holds by applying Lemma 1. This lemma would be applicable because we could obtain a number $N(\bar{s}, a)$ of i.i.d. samples per abstract state-action pair, distributed according to $\bar{T}_{\omega_X}(\cdot|\bar{s}, a)$. However, the samples are not i.i.d. in RLAO: the samples are neither identically distributed nor independent, and this combination means that previous techniques fail. We will first cover the distribution of the samples and show that samples not being identically distributed is not a problem. Then we prove that samples are not guaranteed to be independent. Afterward, Section 3.4 shows that we can still learn when the samples are dependent.

#### 3.3.1 Why We Can Not Use Lemma 1: Dependent Samples.

The samples are not necessarily identically distributed in RLAO since the agent obtains a sample $\bar{Y}^{(i)}$ when taking action $a$ from state $X_{\bar{s},a}^{(i)} = s \in \bar{s}$, as in equation 17. If $X_{\bar{s},a}^{(i)} \neq X_{\bar{s},a}^{(j)}$, these states can have different transition distributions. This implies that in general we might not be able to apply Lemma 1, because it assumes identically distributed random variables. However, different distributions by themselves need not be a problem; we show that the result also holds when the random variables are not identically distributed:

**Lemma 2.** *Let $X_{\bar{s},a} = s_1, \cdots, s_m$ be a sequence of states $s \in \bar{s}$ and let $\bar{Y}_{\bar{s},a} = \bar{Y}^{(1)}, \bar{Y}^{(2)}, \cdots, \bar{Y}^{(m)}$ be independent random variables distributed according to $\Pr(\cdot|s_1, a), \cdots, \Pr(\cdot|s_m, a)$ (equation 7). Then, for all $\epsilon > 0$,*

$$\Pr(||\bar{T}_Y(\cdot|\bar{s}, a) - \bar{T}_{\omega_X}(\cdot|\bar{s}, a)||_1 \geq \epsilon) \leq (2^{|\bar{S}|} - 2)e^{-\frac{1}{2}m\epsilon^2}. \tag{22}$$

The proof can be found in Appendix B. Therefore, if the samples in RLAO were independent, then we could apply Lemma 2 to guarantee an accurate model.

**Independence.** One could be tempted to assume the samples are independent, i.e.,

$$\forall \bar{s}'_1, \cdots, \bar{s}'_m \in (\bar{S})^m : \Pr(\bar{Y}_{\bar{s},a}^{(1)} = \bar{s}'_1, \cdots, \bar{Y}_{\bar{s},a}^{(m)} = \bar{s}'_m) = \Pr(\bar{Y}_{\bar{s},a}^{(1)} = \bar{s}'_1) \cdots P(\bar{Y}_{\bar{s},a}^{(m)} = \bar{s}'_m). \tag{23}$$

However, this is not true in general in RLAO:

**Observation 1.** *When collecting samples online using an abstraction function, such samples are not necessarily independent.*

Samples can be dependent when 1) samples are collected online in the real environment, of which we do not know the transitions, and 2) the samples are collected for abstract states $\bar{s}$. Observation 1 can be understood from the perspective that the RLAO problem corresponds to RL in a POMDP (Bai et al., 2016). The corresponding POMDP uses the abstraction function as the observation function and the abstract states as observations. Since the transitions between observations need not be Markov in POMDPs, the samples from abstract states can depend on the history. While this observation may be clear from the POMDP perspective, work in RLAO regularly assumes (explicitly or implicitly) that independent samples can somehow be obtained (Paduraru et al., 2008; Ortner et al., 2014a; Jiang et al., 2015; Ortner et al., 2019). In the following counterexample, we rigorously show that samples are not necessarily independent.

**Counterexample.** We use the example MDP and abstraction in Figure 2, where we have four states, three abstract states, and only one action. Since the example MDP has only one action, we omit the action from the notation. We examine the transition function of abstract state

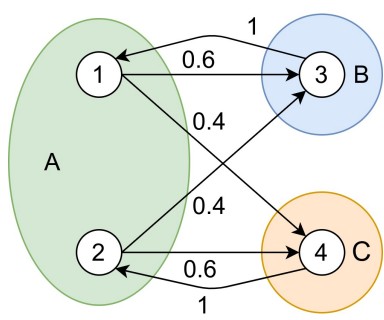

Figure 2: Simple MDP, with only 1 action, and abstraction. The small circles are states (1,2,3,4). A, B and C are the abstract states. The arrows show the transition probabilities, e.g. $P(3|1) = 0.6$.

$A$, $\bar{T}_Y(\cdot|A)$ and consider the first two times we transition from $A$. These two transition samples, $\bar{s}'_1$ and $\bar{s}'_2$, are the first two entries in $\bar{Y}_A$. We show that the samples are not independent for at least one combination of $\bar{s}'_1$ and $\bar{s}'_2$.

Let $\bar{s}'_1 = \bar{s}'_2 = B$, i.e., the first two times we experience a transition from the abstract state $A$, we end up in $B$. We denote the $i$-th experienced transition from abstract state $A$ as $\bar{Y}_A^{(i)}$. Let state 1 be the starting state.

We start with the product of the probabilities:

$$\Pr(\bar{Y}_A^{(1)} = B)\Pr(\bar{Y}_A^{(2)} = B). \tag{24}$$

We have $\Pr(\bar{Y}_A^{(1)} = B) = \Pr(B|1) = 0.6$ for the first term since state 1 is the starting state. The second term is more complex since it includes the probability of starting the transition from state 1 and state 2.

We have:

$$\Pr(\bar{Y}_A^{(2)} = B) = \sum_{\bar{s} \in \bar{S}} \Pr(\bar{Y}_A^{(2)} = B|\bar{Y}_A^{(1)} = \bar{s})\Pr(\bar{Y}_A^{(1)} = \bar{s}) \tag{25}$$

$$= \Pr(\bar{Y}_A^{(2)} = B|\bar{Y}_A^{(1)} = A)\Pr(\bar{Y}_A^{(1)} = A) + \Pr(\bar{Y}_A^{(2)} = B|\bar{Y}_A^{(1)} = B)\Pr(\bar{Y}_A^{(1)} = B)$$

$$+ \Pr(\bar{Y}_A^{(2)} = B|\bar{Y}_A^{(1)} = C)\Pr(\bar{Y}_A^{(1)} = C). \tag{26}$$

$$= \Pr(\bar{Y}_A^{(2)} = B|\bar{Y}_A^{(1)} = B)\Pr(\bar{Y}_A^{(1)} = B) + \Pr(\bar{Y}_A^{(2)} = B|\bar{Y}_A^{(1)} = C)\Pr(\bar{Y}_A^{(1)} = C). \tag{27}$$

$$= \Pr(Y_A^{(2)} = 3|\bar{Y}_A^{(1)} = 3)\Pr(Y_A^{(1)} = 3) + \Pr(Y_A^{(2)} = 3|Y_A^{(1)} = 4)\Pr(Y_A^{(1)} = 4) \tag{28}$$

$$= 0.6 \cdot 0.6 + 0.4 \cdot 0.4 = 0.52. \tag{29}$$

For the step from equation 26 to equation 27, $\Pr(\bar{Y}_A^{(1)} = A)$ is 0 because there is no transition from a state in $A$ to a state in $A$. Then, from equation 27 to equation 28, we use that both abstract states $B$ and $C$ consist of exactly 1 state. So, e.g., $\Pr(\bar{Y}_A^{(2)} = B|\bar{Y}_A^{(1)} = B) = \Pr(Y_A^{(2)} = 3|\bar{Y}_A^{(1)} = 3)$. So, for the product of the probabilities, we end up with: $\Pr(\bar{Y}_A^{(1)} = B)\Pr(\bar{Y}_A^{(2)} = B) = 0.6 \cdot 0.52 = 0.321$.

For the joint probability, we have:

$$Pr(\bar{Y}_A^{(1)} = B, \bar{Y}_A^{(2)} = B) = \Pr(\bar{Y}_A^{(1)} = B)\Pr(\bar{Y}_A^{(2)} = B|\bar{Y}_A^{(1)} = B) \tag{30}$$

$$= \Pr(B|1)(\Pr(B|1)\Pr(1|B)) \tag{31}$$

$$= 0.6 \cdot (0.6 \cdot 1) \tag{32}$$

$$= 0.6 \cdot 0.6 \tag{33}$$

$$= 0.36. \tag{34}$$

Here, $\Pr(\bar{Y}_A^{(2)} = B|\bar{Y}_A^{(1)} = B) = \Pr(B|1)\Pr(1|B)$ because the first transition ends in state $B$ and we always transition to state 1 from state $B$. Hence, $\Pr(\bar{Y}_A^{(2)} = B|\bar{Y}_A^{(1)} = B) = \Pr(B|1)\Pr(1|B) = 0.6 \cdot 1$.

Combining the joint probability and the product of probabilities, we end up with:

$$0.36 = \Pr(\bar{Y}_A^{(1)} = B, \bar{Y}_A^{(2)} = B) \neq \Pr(\bar{Y}_A^{(1)} = B)\Pr(\bar{Y}_A^{(2)} = B) = 0.6 \cdot 0.52. \tag{35}$$

Thus, the samples are not independent. Leading us to the second observation.

**Observation 2.** *As independence cannot be guaranteed, Lemmas 1 and 2 cannot be readily applied to show that the abstract L1 inequality holds.*

This claim follows from the fact that Lemmas 1 and 2 both use the assumption of independence in their proofs. It would still be possible to obtain independent samples if we could, for example, have access to a simulator of the problem. In that case, it is still possible to give guarantees on the accuracy of the model, which we show in Appendix E. However, we consider the setting where a simulator is not available.

### 3.3.2 Why the Approach by Strehl & Littman (2008) Fails

While the counterexample above is informative as to why Lemmas 1 and 2 cannot be applied, the failure to apply these lemmas may not come as a surprise: in the end, as shown by Strehl & Littman (2008), more work is needed. They are able to use these concentration inequalities due to an additional proof that shows that even though the samples are drawn from a Markov chain, and thus not fully independent, the inequality still serves as an upper bound. This raises the question whether we could not follow the same approach, and show that Lemma 1 (or 2) is still an upper bound in the RLAO setting.

It turns out that this is not possible, as that result uses the Markov property and requires each sample to be identically distributed. Without abstraction, only $(s, a)$ and the next states $s' \sim P(\cdot|s, a)$ are considered, which indeed have the same distribution. In RLAO, the outcomes of multiple states are grouped together and for a pair $(\bar{s}, a)$ both the state $s \in \bar{s}$ that we reach and the resulting next state $\bar{s}'$ need to be considered. Since the distributions $s' \sim P(\cdot|s_1, a)$ and $s' \sim P(\cdot|s_2, a)$ of two states $s_1, s_2 \in \bar{s}$ do not have to be the same, these samples are not guaranteed to be identically distributed.

### 3.3.3 Summary: Why Previous Strategies Fail

Summarizing, we have seen that previous strategies fail due to the combination of samples neither being independent, nor being identically distributed. We showed that if the samples would only be non-identically distributed (but independent) we could modify the proof of Lemma 1, leading to Lemma 2, that could be directly used. On the other hand, if the samples were only dependent (but still identically distributed), it would be possible to follow the strategy of Strehl & Littman (2008). However, given that we are dealing with the dependent non-identically distributed setting, neither of these previous strategies work, and a new approach is needed, as we present next.

### 3.4 Guarantees for Abstract Model Learning Using Martingales

Now we want to give a guarantee in the form of the abstract L1 inequality from equation 21.[6] In Section 3.2, we found this was not possible with concentration inequalities such as Hoeffding's inequality because the samples are not guaranteed to be independent. Here we consider a related bound for weakly dependent samples, the Azuma-Hoeffding inequality. This inequality makes use of the properties of a martingale difference sequence, which are slightly weaker than independence:

**Definition 3** (Martingale difference sequence (Azuma, 1967))**.** *The sequence $Z_1, Z_2, \cdots$ is a martingale difference sequence if, $\forall i$, it satisfies the following conditions:*

$$E[Z_i|Z_1, Z_2, \cdots, Z_{i-1}] = 0,$$
$$|Z_i| < \infty.$$

The properties of the martingale difference sequence can be used to obtain the following concentration inequality:

**Lemma 3** (Azuma-Hoeffding Inequality (Hoeffding, 1963; Azuma, 1967))**.** *If the random variables $Z_1, Z_2, \cdots$ form a martingale difference sequence (Def. 3), with $|Z_i| \leq b$, then*

$$\Pr(\sum_{i=1}^{n} Z_i > \epsilon) \leq e^{-\frac{\epsilon^2}{2b^2 n}}. \tag{36}$$

Our main result, Theorem 2, shows that we can use Lemma 3 to obtain a concentration inequality for the abstract transition function in RLAO (as in equation 21). Specifically, we show that, with high probability, the empirical abstract transition function $\bar{T}_Y$ will be close to the abstract transition function $\bar{T}_{\omega_X}$:

---

[6]We focus on the transition function, for the reward function we make some simplifying assumptions in Section 4. We discuss in Section 6 how these assumptions can be relaxed and the result extended for the reward function.

**Theorem 2** (Abstract L1 inequality). *If an agent has access to a state abstraction function $\phi$ and uses this to collect data for any abstract state-action pair $(\bar{s}, a)$ by acting in an MDP $M$ according to a policy $\bar{\pi}$, we have that the following holds with a probability of at least $1 - \delta$ for a fixed value of $N(\bar{s}, a)$:*

$$||\bar{T}_Y(\cdot|\bar{s}, a) - \bar{T}_{\omega_X}(\cdot|\bar{s}, a)||_1 \leq \epsilon, \tag{37}$$

*where $\delta = 2^{|\bar{S}|} e^{-\frac{1}{8} N(\bar{s}, a)\epsilon^2}$.*

Here we use the definitions of $\bar{T}_Y(\cdot|\bar{s}, a)$ (equation 16) and $\bar{T}_{\omega_X}(\cdot|\bar{s}, a)$ (equation 19). This theorem shows that the empirical model constructed by MBRLAO is close to an Abstract MDP $\bar{M}_{\omega_X}$, and here Theorem 1 gives performance loss guarantees. By assuming that $\bar{M}_{\omega_X}$ is the results of an approximate model irrelevance abstraction, we can give end to end guarantees. In simpler words: our result just shows that whatever $\bar{T}_Y$ you might end up with (indeed, regardless of changing policies, etc.), it was generated by some underlying states $X$, and the implied $\bar{T}_{\omega_X}$ will concentrate on $\bar{T}_Y$.

Note that, unlike in planning with abstract MDPs (Section 2.4), there is no fixed set of weights $\bar{T}_{\omega_X}$ that can be used as ground truth that needs to be estimated. As illustrated in Section 3.3.1, the RLAO setting corresponds to a POMDP, which means that *depending on the history* there would be a different distribution over the states (and thus different weights) in each abstract state (called 'the belief' in a POMDP). Instead, both $\bar{T}_{\omega_X}$ and $\bar{T}_Y$ change over time. We show in the proof of Theorem 2 (in Appendix C) that $\bar{T}_Y$ will concentrate on $\bar{T}_{\omega_X}$ as they are intimately connected. This is possible because Lemma 3 can be applied as long as the $Z_i$ form a martingale difference sequence, with $|Z_i| \leq b$. In the proof, we define a suitable $Z_i$ and show that $\bar{T}_Y$ will thus concentrate on $\bar{T}_{\omega_X}$, with high probability.

## 4 An Illustration: R-MAX From Abstracted Observations

Here we give an illustration of how we can use Theorem 2 to provide guarantees for MBRL methods in RLAO with an *approximate model similarity abstraction*. We illustrate this using the R-MAX algorithm (Brafman & Tennenholtz, 2002). We start with a short description of R-MAX and how it operates with abstraction.

The R-MAX algorithm maintains a model of the environment. It uses this model to compute a policy periodically and then follows this policy for several steps. Initially, all the state-action pairs are *unknown* and the algorithm optimistically initializes their reward and transition functions: $R(s, a) = R_{\max}$ (the maximum reward), $T(s|s, a) = 1$, and $\forall s' \neq s : T(s'|s, a) = 0$. This initialization means that, in the model the algorithm maintains, these unknown $(s, a)$ lead to the maximum reward, hence the name R-MAX. A state-action pair's transition and reward function are only updated once they have been visited sufficiently often, at which point the state-action pair is considered *known*. Together, this ensures that the algorithm explores sufficiently. During execution, the algorithm operates in episodes of $n$-steps. At the start of every episode it calculates an optimal $n$-step policy and follows this for $n$ timesteps, or until a state-action pair becomes known. Once all the state-action pairs are known it calculates the optimal policy for the final model and then runs this indefinitely. The algorithm has the following guarantee:

**Theorem 3** (R-MAX in MDPs without abstraction (Brafman & Tennenholtz, 2002)). *Given an MDP $M$, with $|S|$ states and $|A|$ actions, and inputs $\epsilon$ and $\delta$. With probability of at least $1 - \delta$ the R-MAX algorithm will attain an expected average return of $Opt(\prod_M(\epsilon, T_\epsilon)) - 2\epsilon$ within a number of steps polynomial in $|S|, |A|, \frac{1}{\epsilon}\frac{1}{\delta}, T_\epsilon$. Where $T_\epsilon$ is the $\epsilon$-return mixing time of the optimal policy, the policies for $M$ whose $\epsilon$-return mixing time is $T_\epsilon$ are denoted by $\prod_M(\epsilon, T_\epsilon)$, the optimal expected $T_\epsilon$-step undiscounted average return achievable by such policies are denoted by $Opt(\prod_M(\epsilon, T_\epsilon))$.*

Here $T_\epsilon$ is the $\epsilon$-return mixing time of a policy $\pi$, it is the minimum number of steps needed to guarantee that the expected average return is within $\epsilon$ of the optimal expected average return (Kearns & Singh, 2002).

For R-MAX from Abstracted Observations, we make the following assumptions that stem from the original analysis: we assume that the MDP is ergodic (Puterman, 2014), [7] that we know $S$ and $A$, that the reward

---

[7] An ergodic, or recurrent, MDP is an MDP where every state is recurrent under every stationary policy, i.e., asymptotically, every state will be visited infinitely often (Puterman, 2014).

---

**Algorithm 1** Procedure: R-MAX from Abstracted Observations

---

**Input:** $\phi, \delta, \epsilon, T_\epsilon$
**for all** $(\bar{s}, a) \in \bar{S} \times A$ **do**
   $\bar{T}_Y(\bar{s}|\bar{s}, a) = 1$
   $\bar{R}_Y(\bar{s}, a) = R_{\max}$
   $\bar{Y}_{\bar{s}, a} = [\,]$
**end for**
$\bar{M}_Y = \langle \bar{S}, A, \bar{T}_Y, \bar{R}_Y \rangle$
Select $m$, the number of samples required per (abstract) state-action pair to make them known.
// While there is still an unknown state-action pair.
**while** $\min_{(\bar{s}, a)} |Y_{\bar{s}, a}| < m$ **do**
   Compute optimal $T_\epsilon$-step policy $\bar{\pi}$ in $\bar{M}_Y$ for the current abstract state.
   **for** $T_\epsilon$ timesteps **do**
      $\bar{s} = \phi(s)$
      $a = \bar{\pi}(\bar{s})$
      $s', r = \text{Step}(s, a)$
      $s = s'$
      **if** $|\bar{Y}_{\bar{s}, a}| < m$ **then**
         $\bar{Y}_{\bar{s}, a}.\text{append}(\phi(s'))$
         **if** $|\bar{Y}_{\bar{s}, a}| = m$ **then**
            // State-action pair has become known.
            **for all** $\bar{s}' \in \bar{S}$ **do**
               $\bar{T}_Y(\bar{s}'|\bar{s}, a) = \frac{1}{m} \sum_{i=1}^m \mathbb{1}\{\bar{Y}_{\bar{s}, a}^{(i)} = \bar{s}'\}$
            **end for**
            $\bar{R}_Y(\bar{s}, a) = r$
            **break**
         **end if**
      **end if**
   **end for**
**end while**
Compute optimal policy $\bar{\pi}^*$ for $\bar{M}$ and run indefinitely.

---

function is deterministic, and that we know the minimum and maximum reward. W.l.o.g., we assume the rewards are between 0 and $R_{\max}$, with $0 < R_{\max} < \infty$. We add the assumption that the agent has access to an approximate model similarity abstraction function $\phi$ and that each state in an abstract state has the same reward function. [8]

Algorithm 1 shows the procedure for R-MAX from Abstracted Observations. It follows the same steps as the original algorithm, except that it makes use of an abstraction function $\phi$ and maintains an abstract model. As in the original, the input to the algorithm is the allowed failure probability $\delta$, the error bound $\epsilon$, and the $\epsilon$-return mixing time $T_\epsilon$ of an optimal policy. We add the abstraction function $\phi$ as a new input. The algorithm uses this function to observe $\phi(s)$, as in Figure 1, and it builds an empirical (abstract) model from the observations it obtains.

Because the algorithm uses an abstraction function $\phi$, we cannot guarantee the $\epsilon$ error bound. However, with Theorem 4 we can still guarantee an error bound that is a function of $\epsilon$ and the error $\eta$ of the abstraction, thus providing the first finite-sample guarantees for RLAO:

**Theorem 4.** *Given an MDP M, an approximate model similarity abstraction $\phi$, with $\eta_R$ and $\eta_T$, and inputs $|\bar{S}|, |A|, \epsilon, \delta, T_\epsilon$. With probability of at least $1 - \delta$ the R-MAX algorithm adapted to abstraction (Algorithm 1) will attain an expected average return of $Opt(\prod_M(\epsilon, T_\epsilon)) - 3g(\eta_T, \eta_R) - 2\epsilon$ within a number of steps polynomial in $|\bar{S}|, |A|, \frac{1}{\epsilon}\frac{1}{\delta}, T_\epsilon$. Where $T_\epsilon$ is the $\epsilon$-return mixing time of the optimal policy, the policies for M whose $\epsilon$-*

---

[8]Note that this is just a slight simplification as any empirical estimate $\bar{R}$ is guaranteed to be within $\eta_R$ of any $\bar{R}_\omega$, under the assumption that the rewards are deterministic.

*return mixing time is $T_\epsilon$ are denoted by $\prod_M(\epsilon, T_\epsilon)$, the optimal expected $T_\epsilon$-step undiscounted average return achievable by such policies are denoted by $Opt(\prod_M(\epsilon, T_\epsilon))$, and*

$$g(\eta_T, \eta_R) = T_\epsilon \eta_R + \frac{(T_\epsilon - 1)T_\epsilon}{2} \eta_T |\bar{S}|.$$

The proof can be found in Appendix D.2 and follows the line of the original R-MAX proof, using the assumptions mentioned at the start of Section 3. To translate the results to the RLAO setting, we first use the Abstract L1 inequality (Theorem 2) to show that the empirical abstract model is accurate with high probability. Then the performance bounds from Theorem 1 can be used to bound the loss in performance by using an abstract policy based on the empirical abstract model in the MDP $M$ instead of the optimal (ground) policy $\pi^*$. These bounds hold for *any* $\omega_X$ as long as $\omega_X$ is a valid weighting function. That $\omega_X$ will be a valid weighting function follows from its definition in equation 18. Because Theorem 1 allows us to directly bound the loss in performance for using an abstract policy, based on an abstract empirical model, in the original problem $M$, the amount of steps is polynomial only in $|\bar{S}|$ instead of $|S|$.

As is typical with abstraction, there is a trade-off between the performance and the required number of steps: a coarser abstract model can potentially learn much faster but could sacrifice optimality, while a non-abstract model might have the best performance in the limit of infinite experience. We can see this trade-off in the results of Theorems 3 and 4. When we directly model $M$ without abstraction, Theorem 3 shows that the algorithm will attain an expected return of $Opt(\prod_M(\epsilon, T_\epsilon)) - 2\epsilon$ within a number of steps polynomial in $|S|, |A|, \frac{1}{\epsilon} \frac{1}{\delta}, T_\epsilon$. Theorem 4 shows that, when we use an approximate model similarity abstraction to learn an abstract model, this leads to an additional performance loss of $3g(\eta_T, \eta_R)$ due to the approximation. However, the advantage of using the abstraction is that the number of steps within which this is achieved is polynomial only in the size of the abstract space $|\bar{S}|$ rather than the (larger) original state space $|S|$. Thus, these results show that the performance is not arbitrarily bad with approximate model similarity abstraction. Moreover, when the abstraction errors ($\eta_T$ and $\eta_R$) are small and the reduction in state space is large, abstraction helps to reach near-optimal performance significantly faster.

## 5 Related Work

The problem we resolved in this paper may seem intuitive, but as we will make clear here, it is a fundamental problem in rich literature. Many studies have considered the combination of abstraction with either planning or RL. Some studies avoid, or ignore, the issue of dependency by simply assuming that samples are independent (Paduraru et al., 2008; Ortner et al., 2014b; Jiang et al., 2015; Badings et al., 2022). Others avoid it by looking at convergence in the limit (Singh et al., 1995; Hutter, 2016; Majeed & Hutter, 2018) or by assuming access to an MDP model (Hallak et al., 2013; Maillard et al., 2013; Ortner et al., 2019).

**RL With Abstraction.** A negative result has been provided in the RLAO setting, showing that R-MAX (Brafman & Tennenholtz, 2002) no longer maintains its guarantees when paired with any state abstraction function (Abel et al., 2018). For this negative result, they give an example that uses approximate $Q^*$ similarity abstractions (Abel et al., 2016). Our counterexample is more powerful: indicating problems with the analysis even for approximate model similarity abstractions (an approximate model similarity abstraction is also an approximate $Q^*$ abstraction, but not the other way around). Nevertheless, our second result shows that it is still possible to give guarantees in RLAO for R-MAX-like algorithms when we use an approximate model similarity abstraction and take the $\eta_R$ and $\eta_T$ inaccuracies into account.

Another study considered a setting related to abstraction, where the transition and reward functions can change over time, either abruptly or gradually (Ortner et al., 2020). The reward and transition probabilities depend on the timestep $t$, so $T(s'|s, a, t)$ instead of $T(s'|s, a)$. They bound the variation in the reward and transition functions over time. By taking the variation over time into account they are able to give results. In their setting, the MDP is fixed given the timestep. However, in RLAO this is not fixed. Each time the transition function at a timestep $t$ could be different.

Recently regret guarantees have been found for the episodic (continuous) RL setting (Sinclair et al., 2022). Their abstraction method learns an abstraction adaptively. The model-based version of their algorithm

requires that the state and actions spaces are embedded in compact metric spaces. In this way, they can define a measure of the difference between state-action pairs in these metric spaces. However, they require oracle access to this metric. In our setting this would require knowing the transition and reward functions, which we assume we do not.

**Abstraction Selection.** There are quite a few studies in the area of *abstraction selection*, where the agent has access to a set of abstraction functions (state representations)(Hallak et al., 2013; Maillard et al., 2013; Lattimore et al., 2013; Ortner et al., 2014a; 2019). Several studies assume that the given set of state representations contains at least one Markov model (Hallak et al., 2013; Maillard et al., 2013; Ortner et al., 2019). One study gives asymptotic guarantees for selecting the correct model and building an exact MDP model (Hallak et al., 2013). The assumption that an MDP model exists in the given set of representations is crucial in their analysis since the samples are i.i.d. for this MDP model. Similarly, other studies also assume that the given set of state representations contains a Markov model (Maillard et al., 2013; Ortner et al., 2019). They create an algorithm for which they obtain regret bounds, and their analysis also uses the Markov representation.

Some studies in abstraction selection do not assume the given abstraction functions contain a Markov model (Lattimore et al., 2013; Ortner et al., 2014a). Lattimore et al. (2013) deal with a more general setting where the problem may be non-Markovian. Instead of assuming access to a set of abstractions, they assume access to a set of models, including a model of the true environment. Since they are given models, they do not focus on learning them, making it very different from our setting. By observing the rewards obtained while executing a policy they are able to exclude unlikely models, and eventually find the true model of the environment. The other study (Ortner et al., 2014a) uses Theorem 2.1 from Weissman et al. (2003), which requires i.i.d. samples. We have shown that independent samples *cannot* be guaranteed in RLAO.

**MDPs With Rich Observations.** Other related work is in MDPs with rich observations or block structure (Azizzadenesheli et al., 2016; Du et al., 2019; Allen et al., 2021). In that setting, each observation can only be generated from a *single* hidden state, which means that the issue of non-i.i.d. data due to abstraction does not arise. We can view the rich observation setting as an aggregation problem, where the observations can be aggregated to form a small (latent) MDP (Azizzadenesheli et al., 2016). Their setting is related to exact model similarity (or bisimulation) (Du et al., 2019). In contrast, in RLAO, each observation can be generated from multiple hidden states, and we do not try to learn the MDP, as it is not small. Furthermore, we focus on approximate model similarity, which introduces the problems as described in Section 3.2.

**I.I.D. Samples.** One way to avoid the issue of dependent samples is by assuming that samples are obtained independently (Paduraru et al., 2008; Ortner et al., 2014b; Jiang et al., 2015; Badings et al., 2022). One study considers the setting with a continuous domain where we are given a data set with i.i.d. samples (Paduraru et al., 2008). They use discretization to aggregate states into abstract states and give a guarantee that, with a high probability, the model will be $\epsilon$-accurate given a fixed data set. While they assume that the data has been gathered i.i.d., our results show that martingale concentration inequalities could be used to extend their results to the online data collection in the RLAO setting. Discretization has been used in another study in a continuous space (Badings et al., 2022). They search for a solution for a linear dynamical system, where the transitions are deterministic, except for an additive noise component. They assume this noise is distributed i.i.d. and try to learn the resulting abstract transition functions by iteratively sampling $N$ samples per abstract-state action pair until a threshold is reached. They assume that samples can be obtained cheaply, e.g., through a simulator, whereas we focus on the exploration problem where data has to be collected online and is expensive. Another study operates in the abstraction selection setting (Jiang et al., 2015). While they do not assume that a Markov model exists in the given set of abstraction functions, they assume a given data set, with i.i.d. data. They give a bound on how accurate the Q-values based on the (implicitly) learned model will be rather than on the accuracy of the model itself. As we showed, the assumption that the data is i.i.d. is not trivial since it means the data cannot just be collected online. Another study's primary focus is on bandits but also gives results for MDPs with a coloring function (Ortner et al., 2014b). We can view state aggregation as a special case of this coloring. They extend the results from UCRL2 (Jaksch et al.,

2010) to the setting with a coloring function. They use the Azuma-Hoeffding inequality for the transition function, which holds for weakly dependent samples. However, they assume the samples are independent and do not show the martingale difference sequence property for the (actually dependent) samples.

**Asymptotic Results.** Another way to deal with dependence between samples is by looking at convergence in the limit (Singh et al., 1995; Hutter, 2016; Majeed & Hutter, 2018). One study gives an asymptotic result for convergence of Q-learning and TD(0) in MDPs with soft state aggregation (Singh et al., 1995). In soft state aggregation, a state $s$ belongs to a cluster $x$ with some probability $P(x|s)$, which means a state $s$ can belong to several clusters. Their result requires an ergodic MDP and a stationary policy that assigns a non-zero probability to every action. Together these imply a limiting state distribution, and they use this to show convergence asymptotically. Another study gives multiple results focusing on approximate and exact abstractions in environments without MDP assumptions (Hutter, 2016). Several of these results are in the planning setting, similar to other planning results for approximate abstractions (Abel et al., 2016). Most relevant for us is their Theorem 12, which for online RL shows convergence in the limit of the empirical transition function under weak conditions, e.g., when the abstract process is an MDP. Under this condition, however, the problem reduces to RL in an (abstract) MDP rather than RLAO. Follow-up work builds on some of these results and focuses on the combination of model-free RL and exact abstraction, also without MDP assumptions (Majeed & Hutter, 2018). They define and operate in a Q-Value Uniform Decision Process, with a mapping from histories to (non-Markovian) states and a "state-uniformity condition". The state-uniformity means that if two histories map to the same state $s$, their optimal Q-values are also the same. They show that, under state-uniformity, Q-learning converges in the limit to the optimal solution. In contrast to our setting, they used an exact abstraction and left extending the results to approximate abstraction as an open question.

**Planning and Abstraction.** For planning in abstract MDPs, there are results for exact state abstractions (Li et al., 2006) and approximate state abstractions (Abel et al., 2016). The results for approximate state abstractions allow for quantifying an upper bound on performance for the optimal policy of an abstract MDP, e.g., as in Theorem 1 for approximate model similarity in Section 2.3. A study built on these results by giving a result for performing RL interacting with an explicitly constructed abstract MDP (Taïga et al., 2018); since the abstract MDP is still an MDP, this is different from RLAO.

**MBRL Using I.I.D. Bounds.** Concentration inequalities for i.i.d. samples, such as the result from Weissman et al. (2003), are often directly applied to the empirical transition function (Brafman & Tennenholtz, 2002; Jaksch et al., 2010; Fruit et al., 2018; Bourel et al., 2020), without mentioning that these samples in a simple RL trajectory may not be independent as shown for instance by Strehl & Littman (2008) in a non-communicating MDP. [9] Strehl & Littman (2008) show that there dependence is not a problem because it is still possible to use a concentration inequality for independent samples, e.g., Hoeffding's inequality, as an upper bound, which implies that derived performance loss bounds are valid. However, their proof uses that transitions and rewards are identically distributed, which is not guaranteed in RLAO.

**RL Using Martingale Bounds.** Martingale concentration inequalities have been used regularly in online RL analysis (Strehl & Littman, 2008; Li, 2009; Jaksch et al., 2010; Lattimore et al., 2013; Maillard et al., 2013; Ortner et al., 2014b; Azizzadenesheli et al., 2016; Ortner et al., 2019; 2020). Our novelty is in using it in RLAO, where we use it to show that we can learn an accurate model and provide performance guarantees in this setting. Several works that employ martingale concentration inequalities are not in the RLAO setting or do not use them for the transition model, and instead apply them to other parts of the analysis such as bounding the difference between the actual and expected returns (Strehl & Littman, 2008; Jaksch et al., 2010; Lattimore et al., 2013; Maillard et al., 2013; Ortner et al., 2014a; 2019). Other works do use martingales for a transition model (Li, 2009; Ortner et al., 2014b; Azizzadenesheli et al., 2016; Ortner et al., 2020). However, these either (implicitly or explicitly) assume samples to be independent (Ortner et al., 2014b; 2020) or identically distributed (Li, 2009; Azizzadenesheli et al., 2016), unlike our analysis. Since, as we

---

[9]An MDP is communicating if, for all $s_1, s_2 \in S$, a deterministic policy exists that eventually leads from $s_1$ to $s_2$ (Puterman, 2014).

detailed in Section 3, independent nor identically distributed samples cannot be guaranteed in RLAO, their analyses do not extend to this setting.

## 6  Discussion and Future Work

Some assumptions we made, i.e., that the reward function is deterministic and each state in an abstract state has the same reward function, can be relaxed. To accurately learn an abstract reward function, one should define a suitable martingale difference sequence, after which Lemma 3 can be used. We considered approximate model similarity abstraction and used the properties of this abstraction to establish an upper bound on the difference in value between the original MDP and an abstract MDP under any abstract policy. This bound was imperative for our results. We established this bound by proof of induction on the difference in value for a horizon $n$. This technique could be used to establish similar bounds and extend our results for other abstractions, e.g., approximate $Q^*$ similarity abstractions (Abel et al., 2016).

Our analysis showed how to extend the results of R-MAX (Brafman & Tennenholtz, 2002) to RLAO. Extending results of other algorithms, e.g., MBIE (Strehl & Littman, 2008) and UCRL2 (Jaksch et al., 2010), requires adapting to slightly different assumptions. For instance, R-MAX assumes ergodicity, while UCRL2 and MBIE assume the problem is communicating and non-communicating, respectively. Other algorithms sometimes use concentration inequalities other than Hoeffding's Inequality, e.g., the empirical Bernstein inequality (Audibert et al., 2007; Maurer & Pontil, 2009) or the Chernoff bound. To adapt these, we could, for instance, use Bernstein-type inequalities for martingales (Dzhaparidze & Van Zanten, 2001).

Theorem 4 shows that, despite problems with dependence, we can give finite-sample guarantees when combining approximate model similarity abstractions with MBRL. For good abstraction functions, i.e., when $\eta_R$ and $\eta_T$ are small and $|S| \gg |\bar{S}|$, this leads to near-optimal solutions while needing fewer samples, compared to learning without abstraction. Practically, for tabular methods, these results mostly mean that concentration inequalities for independent samples have to be replaced in RLAO, for example by concentration inequalities based on martingales, as we have shown here. In deep model-based RL, several recent empirical works have shown promising results by focusing on learning exact abstractions (Biza & Platt, 2019; van der Pol et al., 2020). An interesting direction is adapting these methods to learn approximate abstractions instead of exact abstractions. Since, compared to exact model similarity abstractions, approximate model similarity abstraction generally results in a smaller (abstract) state space; this could lead to faster learning.

Our results shed further light on the observation from Abel et al. (2018) that RLAO is different from performing RL in an MDP constructed with abstraction. As our observations show, in RLAO the transition functions are not static, the samples are not identically distributed, and cannot be guaranteed to be independent. This could mean that in situations where we want to learn an abstraction, the behavior is also not quite as expected. In such situations, similar approaches that we applied here may prove useful, as many situations in RL have already been shown to not be independent processes. While our results hold for approximate model similarity models, there could be even more compact representations for which our techniques could lead to similar results. One clear example would be abstractions that focus not on state abstraction, but rather on state-action abstraction, of which state abstraction is simply a special case.

People have been applying MDPs and RL to all kinds of problems, even though we know that the Markov property very rarely holds. Given that almost all theory of RL critically depends on this property, one could wonder why these things even work? Intuitively, we expect that the states in these successful applications are somehow "Markovian enough". In this work, we provide an understanding of this vague concept. Specifically, we show that an existing criterion of state representations (approximate model similarity) in fact is a formal notion of "Markovian enough" in MBRL. Thus, it provides critical insight into under what circumstances (and therefore in what applications) MBRL methods are expected to work.

## 7  Conclusion

We analyzed RLAO: online MBRL combined with state abstraction when the model of the MDP is unavailable. Via a counterexample, we showed that it cannot be guaranteed that samples obtained online in RLAO

are independent. Many current guarantees from MBRL methods use concentration results that assume i.i.d. samples, e.g., Theorem 2.1 from Weissman et al. (2003), the empirical Bernstein inequality (Audibert et al., 2007; Maurer & Pontil, 2009), or the Chernoff bound. Because they use these concentration inequalities, their guarantees do not hold in RLAO. In fact, none of the existing analyses of MBRL apply to RLAO. We showed that samples in RLAO are only weakly dependent and that concentration inequalities for (weakly) dependent variables, such as Lemma 3, are a viable alternative through which we can come to guarantees on the empirical model. We used this result to present the first sample efficient learning results for RLAO, thus showing it is possible to combine the benefits of abstraction and MBRL. These results showcase under what circumstances performance guarantees in MBRL can be transferred to settings with abstraction.

### Acknowledgments

This project received funding from the European Research Council (ERC) under the European Union's Horizon 2020 research and innovation programme (grant agreement No. 758824 —INFLUENCE).

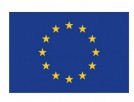 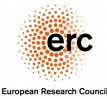

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

## A  Well Known Results

We restate some well-known results that we use in the proofs in the other sections.

### A.1  Hoeffding's Inequality

Hoeffding's inequality can tell us the probability that the average of $m$ random independent (but not necessarily identically distributed) samples deviates more than $\epsilon$ from its expectation.

Let $Z^{(1)}, Z^{(2)}, \cdots, Z^{(m)}$ be bounded independent random variables, and let $\bar{Z}$ and $\mu$ be defined as

$$\bar{Z} \triangleq \frac{Z^{(1)} + \cdots + Z^{(m)}}{m}, \tag{38}$$

$$\mu \triangleq E[\bar{Z}] = \frac{E[Z^{(1)} + \cdots + Z^{(m)}]}{m}. \tag{39}$$

Then Hoeffding's inequality states:

**Lemma 4** (Hoeffding's inequality (Hoeffding, 1963)). *If $Z^{(1)}, Z^{(2)}, \cdots, Z^{(m)}$ are independent and $0 \leq Z^{(i)} \leq 1$ for $i = 1, \cdots, m$, then for $0 < \epsilon < 1 - \mu$ we have the following inequalities*

$$\Pr(\bar{Z} - \mu \geq \epsilon) \leq e^{-2m\epsilon^2}, \tag{40}$$

$$\Pr(|\bar{Z} - \mu| \geq \epsilon) \leq 2e^{-2m\epsilon^2}, \tag{41}$$

$$\Pr(\sum_{i=1}^{m}(Z^{(i)} - \mu) \geq \epsilon) \leq e^{-2\frac{\epsilon^2}{m}}, \tag{42}$$

$$\Pr(|\sum_{i=1}^{m}(Z^{(i)} - \mu)| \geq \epsilon) \leq 2e^{-2\frac{\epsilon^2}{m}}. \tag{43}$$

### A.2  Union Bound

Given that we have a set of events, the union bound allows us to upper bound the probability that at least one of the events happens, even when these events are not independent.

**Lemma 5** (Union Bound (Boole, 1854)). *For a countable set of events $A_1, A_2, A_3, \cdots$, we have*

$$\Pr(\cup_i A_i) \leq \sum_i \Pr(A_i). \tag{44}$$

I.e., the probability that at least one of the events happens is, at most, the sum of the probabilities of the individual events.

### A.3  Value Bounds for Abstract and True Models

Here we give upper bounds on the difference in value between the real MDP and an abstract MDP under various policies. We will use these bounds in Appendix D to adapt the results of R-MAX (Brafman & Tennenholtz, 2002) to RLAO. These bounds and proofs are very similar to existing bounds (Brafman & Tennenholtz, 2002; Strehl & Littman, 2008; Abel et al., 2016; Taïga et al., 2018). Here we repeat these for abstract models in the undiscounted finite horizon and in the discounted infinite horizon.

We define the finite horizon value function $\forall s \in S$:

$$V^{\pi,n}(s) = R(s, \pi(s)) + \sum_{s' \in S} T(s'|s, \pi(s))V^{\pi,n-1}(s), \tag{45}$$

$$V^{\pi,1}(s) = R(s, \pi(s)). \tag{46}$$

We use $V^{\bar{\pi},n}$ to denote the value in $M$ under policy $\bar{\pi}$ and $\bar{V}^{\bar{\pi},n}$ to denote the value in $\bar{M}$ under policy $\bar{\pi}$.

**Theorem 1.** *Let $M = \langle S, A, T, R \rangle$ be an MDP and $\bar{M} = \langle \bar{S}, A, \bar{T}, \bar{R} \rangle$ an abstract MDP, for some defined abstract transitions and rewards. We assume that*

$$\forall \bar{s}, \bar{s}' \in \bar{S}, \ s \in \bar{s}, \ a \in A : |\bar{T}(\bar{s}'|\bar{s}, a) - \Pr(\bar{s}'|s, a)| \leq \eta_T$$
$$and \ |\bar{R}(\bar{s}, a) - R(s, a)| \leq \eta_R. \tag{47}$$

*Then, for a finite horizon problem with horizon $h$ we have:*

$$V^*(s) - V^{\bar{\pi}^*}(s) \leq 2h\eta_R + (h+1)h\eta_T|\bar{S}|R_{max}. \tag{48}$$

*And for a discounted infinite horizon problem with discount $\gamma$ we have:*

$$V^*(s) - V^{\bar{\pi}^*}(s) \leq \frac{2\eta_R}{1-\gamma} + \frac{2\gamma\eta_T|\bar{S}|R_{max}}{(1-\gamma)^2}. \tag{49}$$

We will use the following two Lemmas to proof the Theorem.

**Lemma 6.** *Under the assumption of equation 47 and for every abstract policy $\bar{\pi}$ and for every state $s \in \bar{s}$, we have: for a finite horizon problem with horizon $h$:*

$$|V^{\bar{\pi},h}(s) - \bar{V}^{\bar{\pi},h}(s)| \leq h\eta_R + \frac{(h-1)h}{2}\eta_T|\bar{S}|R_{max}, \tag{50}$$

*and for a discounted infinite horizon problem with discount $\gamma$:*

$$|V^{\bar{\pi}}(s) - \bar{V}^{\bar{\pi}}(s)| \leq \frac{\eta_R}{1-\gamma} + \frac{\gamma\eta_T|\bar{S}|R_{max}}{(1-\gamma)^2}. \tag{51}$$

*Proof.* The proof follows the same steps for both the discounted infinite horizon and the undiscounted finite horizon. For completeness, we show them both here.

First, for the undiscounted finite horizon. By induction, we will show that for $n \geq 1$

$$\forall \bar{s} \in \bar{S}, s \in \bar{s} : |V^{\bar{\pi},n}(s) - \bar{V}^{\bar{\pi},n}(s)| \leq n\eta_R + \frac{(n-1)n}{2}\eta_T|\bar{S}|R_{\max}. \tag{52}$$

For $n = 1$, we have

$$|V^{\bar{\pi},1}(s) - \bar{V}^{\bar{\pi},1}(s)| = |R(s, \pi(\bar{s})) - \bar{R}(\bar{s}, \bar{\pi}(\bar{s}))| \leq \eta_R. \tag{53}$$

Now assume that the induction hypothesis, equation 52, holds for $n - 1$, then

$$|V^{\bar{\pi},n}(s) - \bar{V}^{\bar{\pi},n}(s)| = |R(s, \bar{\pi}(\bar{s})) - \bar{R}(\bar{s}, \bar{\pi}(\bar{s})) + \sum_{s' \in S} T(s'|s, \bar{\pi}(s))V^{\bar{\pi},n-1}(s') - \sum_{\bar{s}' \in \bar{S}} \bar{T}(\bar{s}'|\bar{s}, \bar{\pi}(\bar{s}))\bar{V}^{\bar{\pi},n-1}(\bar{s}')| \tag{54}$$

$$\leq |R(s, \bar{\pi}(\bar{s})) - \bar{R}(\bar{s}, \bar{\pi}(\bar{s}))| + |\sum_{\bar{s}' \in \bar{S}} \sum_{s' \in \bar{s}'} T(s'|s, \bar{\pi}(s))V^{\bar{\pi},n-1}(s') - \sum_{\bar{s}' \in \bar{S}} \bar{T}(\bar{s}'|\bar{s}, \bar{\pi}(\bar{s}))\bar{V}^{\bar{\pi},n-1}(\bar{s}')| \tag{55}$$

$$\leq \eta_R + |\sum_{\bar{s}' \in \bar{S}} \sum_{s' \in \bar{s}'} T(s'|s, \bar{\pi}(s))V^{\bar{\pi},n-1}(s') - \sum_{\bar{s}' \in \bar{S}} \bar{V}^{\bar{\pi},n-1}(\bar{s}') \sum_{s' \in \bar{s}'} T(s'|s, \bar{\pi}(\bar{s}))|$$
$$+ |\sum_{\bar{s}' \in \bar{S}} \bar{V}^{\bar{\pi},n-1}(\bar{s}') \sum_{s' \in \bar{s}'} T(s'|s, \bar{\pi}(\bar{s})) - \sum_{\bar{s}' \in \bar{S}} \bar{T}(\bar{s}'|\bar{s}, \bar{\pi}(\bar{s}))\bar{V}^{\bar{\pi},n-1}(\bar{s}')| \tag{56}$$

$$\leq \eta_R + |\sum_{\bar{s}' \in \bar{S}} \sum_{s' \in \bar{s}'} T(s'|s, \bar{\pi}(s))[V^{\bar{\pi},n-1}(s') - \bar{V}^{\bar{\pi},n-1}(\bar{s}')]| + |\sum_{\bar{s}' \in \bar{S}} [\bar{T}(\bar{s}'|\bar{s}, \bar{\pi}(\bar{s})) - \sum_{s' \in \bar{s}'} T(s'|s, \bar{\pi}(\bar{s}))]\bar{V}^{\bar{\pi},n-1}(\bar{s}')| \tag{57}$$

$$\leq \eta_R + (n-1)\eta_R + \frac{(n-1-1)(n-1)}{2}\eta_T|\bar{S}|R_{\max} + \eta_T|\bar{S}|(n-1)R_{\max} \tag{58}$$

$$= n\eta_R + \frac{(n-2)(n-1)}{2}\eta_T|\bar{S}|R_{\max} + \eta_T|\bar{S}|(n-1)R_{\max} \tag{59}$$

$$= n\eta_R + \frac{(n-1)n}{2}\eta_T|\bar{S}|R_{\max}. \tag{60}$$

For the step from equation 55 to equation 56, we subtract and add $\sum_{\bar{s}' \in \bar{S}} \bar{V}^{\bar{\pi},n-1}(\bar{s}') \sum_{s' \in \bar{s}'} T(s'|s,\bar{\pi}(\bar{s}))$, and from equation 57 to equation 58, we use the inductive hypothesis and the fact that $(n-1)R_{\max}$ is an upper bound on $\bar{V}^{\bar{\pi},n-1}(\bar{s}')$ since the maximum reward per timestep is $R_{\max}$.

Now, for the discounted infinite horizon. By induction, we will show that for $n \geq 1$

$$\forall \bar{s} \in \bar{S}, s \in \bar{s} : |V^{\bar{\pi},n}(s) - \bar{V}^{\bar{\pi},n}(s)| \leq \eta_R \gamma^{n-1} + \sum_{i=0}^{n-2} \gamma^i (\eta_R + \frac{\gamma \eta_T |\bar{S}| R_{\max}}{1-\gamma}). \tag{61}$$

For $n = 1$, we have

$$|V^{\bar{\pi},1}(s) - \bar{V}^{\bar{\pi},1}(s)| = |R(s,\pi(\bar{s})) - \bar{R}(\bar{s},\bar{\pi}(\bar{s}))| \leq \eta_R. \tag{62}$$

Now assume that the induction hypothesis, equation 61, holds for $n-1$, then

$$|V^{\bar{\pi},n}(s) - \bar{V}^{\bar{\pi},n}(s)| = |R(s,\bar{\pi}(\bar{s})) - \bar{R}(\bar{s},\bar{\pi}(\bar{s})) + \gamma(\sum_{s' \in S} T(s'|s,\bar{\pi}(s))V^{\bar{\pi},n-1}(s') - \sum_{\bar{s}' \in \bar{S}} \bar{T}(\bar{s}'|\bar{s},\bar{\pi}(\bar{s}))\bar{V}^{\bar{\pi},n-1}(\bar{s}'))| \tag{63}$$

$$\leq |R(s,\bar{\pi}(\bar{s})) - \bar{R}(\bar{s},\bar{\pi}(\bar{s}))| + \gamma| \sum_{\bar{s}' \in \bar{S}} \sum_{s' \in \bar{s}'} T(s'|s,\bar{\pi}(s))V^{\bar{\pi},n-1}(s') - \sum_{\bar{s}' \in \bar{S}} \bar{T}(\bar{s}'|\bar{s},\bar{\pi}(\bar{s}))\bar{V}^{\bar{\pi},n-1}(\bar{s}')| \tag{64}$$

$$\leq \eta_R + \gamma| \sum_{\bar{s}' \in \bar{S}} \sum_{s' \in \bar{s}'} T(s'|s,\bar{\pi}(s))V^{\bar{\pi},n-1}(s') - \sum_{\bar{s}' \in \bar{S}} \bar{V}^{\bar{\pi},n-1}(\bar{s}') \sum_{s' \in \bar{s}'} T(s'|s,\bar{\pi}(\bar{s}))|$$
$$+ \gamma| \sum_{\bar{s}' \in \bar{S}} \bar{V}^{\bar{\pi},n-1}(\bar{s}') \sum_{s' \in \bar{s}'} T(s'|s,\bar{\pi}(\bar{s})) - \sum_{\bar{s}' \in \bar{S}} \bar{T}(\bar{s}'|\bar{s},\bar{\pi}(\bar{s}))\bar{V}^{\bar{\pi},n-1}(\bar{s}')| \tag{65}$$

$$\leq \eta_R + \gamma| \sum_{\bar{s}' \in \bar{S}} \sum_{s' \in \bar{s}'} T(s'|s,\bar{\pi}(s)) \big[ V^{\bar{\pi},n-1}(s') - \bar{V}^{\bar{\pi},n-1}(\bar{s}') \big]| + \gamma| \sum_{\bar{s}' \in \bar{S}} \big[ \bar{T}(\bar{s}'|\bar{s},\bar{\pi}(\bar{s})) - \sum_{s' \in \bar{s}'} T(s'|s,\bar{\pi}(\bar{s})) \big] \bar{V}^{\bar{\pi},n-1}(\bar{s}')| \tag{66}$$

$$\leq \eta_R + \gamma(\eta_R \gamma^{n-2} + \sum_{i=0}^{n-3} \gamma^i (\eta_R + \frac{\gamma \eta_T |\bar{S}| R_{\max}}{1-\gamma})) + \gamma \eta_T |\bar{S}| \frac{R_{\max}}{1-\gamma} \tag{67}$$

$$= \eta_R + \eta_R \gamma^{n-1} + \sum_{i=1}^{n-2} \gamma^i (\eta_R + \frac{\gamma \eta_T |\bar{S}| R_{\max}}{1-\gamma}) + \gamma \eta_T |\bar{S}| \frac{R_{\max}}{1-\gamma} \tag{68}$$

$$= \eta_R \gamma^{n-1} + \sum_{i=0}^{n-2} \gamma^i (\eta_R + \frac{\gamma \eta_T |\bar{S}| R_{\max}}{1-\gamma}). \tag{69}$$

For the step from equation 64 to equation 65, we subtract and add $\sum_{\bar{s}' \in \bar{S}} \bar{V}^{\bar{\pi},n-1}(\bar{s}') \sum_{s' \in \bar{s}'} T(s'|s,\bar{\pi}(\bar{s}))$, and from equation 66 to equation 67, we use the inductive hypothesis and the fact that $\frac{R_{\max}}{1-\gamma}$ is an upper bound on $\bar{V}^{\bar{\pi},n-1}(\bar{s}')$.

Finally, taking the limit for $n \to \infty$, we get:

$$|V^{\bar{\pi}}(s) - \bar{V}^{\bar{\pi}}(s)| \leq \eta_R \times 0 + \frac{1}{1-\gamma}(\eta_R + \frac{\gamma \eta_T |\bar{S}| R_{\max}}{1-\gamma})$$
$$= \frac{\eta_R}{1-\gamma} + \frac{\gamma \eta_T |\bar{S}| R_{\max}}{(1-\gamma)^2}.$$

$\square$

**Lemma 7.** *Under the assumption of equation 47 and for every state $s \in \bar{s}$, we have: for a finite horizon problem with horizon $h$:*

$$|V^{*,h}(s) - \bar{V}^{*,h}(s)| \leq h\eta_R + \frac{(h-1)h}{2}\eta_T |\bar{S}| R_{max}, \tag{70}$$

*and for a discounted infinite horizon problem with discount $\gamma$:*

$$|V^*(s) - \bar{V}^*(s)| \leq \frac{\eta_R}{1-\gamma} + \frac{\gamma\eta_T|\bar{S}|R_{max}}{(1-\gamma)^2}. \tag{71}$$

*Proof.* First, we define

$$\forall \bar{s} \in \bar{S}, s \in S : V^{*,n}(s) = \max_{a \in A}\left[R(s,a) + \gamma \sum_{s' \in S} T(s'|s,a)V^{*,n-1}(s')\right], \tag{72}$$

$$\bar{V}^{*,n}(\bar{s}) = \max_{a \in A}\left[\bar{R}(\bar{s},a) + \gamma \sum_{\bar{s}' \in \bar{S}} \bar{T}(\bar{s}'|\bar{s},a)\bar{V}^{*,n-1}(\bar{s}')\right]. \tag{73}$$

For the undiscounted case $\gamma = 1$, so we can drop $\gamma$ from the notation.

The proof follows the same steps as the proof of Lemma 6. We start again with the undiscounted finite horizon.

By induction, we will show that for $n \geq 1$

$$\forall \bar{s} \in \bar{S}, s \in \bar{s} : |V^{*,n}(s) - \bar{V}^{*,n}(\bar{s})| \leq n\eta_R + \frac{(n-1)n}{2}\eta_T|\bar{S}|R_{\max}. \tag{74}$$

Making use of the fact that $|\max f - \max g| \leq \max|f - g|$, we have for $n = 1$

$$|V^{*,1}(s) - \bar{V}^{*,1}(\bar{s})| = |\max_{a \in A} R(s,a) - \max_{a \in A}\bar{R}(\bar{s},a)| \leq \max_{a \in A}|R(s,a) - \bar{R}(\bar{s},a)| \leq \eta_R. \tag{75}$$

Now assume that the induction hypothesis, equation 74, holds for $n-1$, then

$$|V^{*,n}(s) - \bar{V}^{*,n}(\bar{s})| = \max_{a \in A}|R(s,a) - \bar{R}(\bar{s},a) + \sum_{s' \in S} T(s'|s,a)V^{*,n-1}(s') - \sum_{\bar{s}' \in \bar{S}} \bar{T}(\bar{s}'|\bar{s},a)\bar{V}^{*,n-1}(\bar{s}')| \tag{76}$$

$$\leq \max_{a \in A}|R(s,a) - \bar{R}(\bar{s},a)| + \max_{a \in A}|\sum_{\bar{s}' \in \bar{S}}\sum_{s' \in \bar{s}'} T(s'|s,a)V^{*,n-1}(s') - \sum_{\bar{s}' \in \bar{S}} \bar{T}(\bar{s}'|\bar{s},a)\bar{V}^{*,n-1}(\bar{s}')| \tag{77}$$

$$\leq \eta_R + \max_{a \in A}|\sum_{\bar{s}' \in \bar{S}}\sum_{s' \in \bar{s}'} T(s'|s,a)V^{*,n-1}(s') - \sum_{\bar{s}' \in \bar{S}} \bar{V}^{*,n-1}(\bar{s}') \sum_{s' \in \bar{s}'} T(s'|s,a)|$$

$$+ \max_{a \in A}|\sum_{\bar{s}' \in \bar{S}} \bar{V}^{*,n-1}(\bar{s}') \sum_{s' \in \bar{s}'} T(s'|s,a) - \sum_{\bar{s}' \in \bar{S}} \bar{T}(\bar{s}'|\bar{s},a)\bar{V}^{*,n-1}(\bar{s}')| \tag{78}$$

$$\leq \eta_R + \max_{a \in A}|\sum_{\bar{s}' \in \bar{S}}\sum_{s' \in \bar{s}'} T(s'|s,a)\left[V^{*,n-1}(s') - \bar{V}^{*,n-1}(\bar{s}')\right]| + \max_{a \in A}|\sum_{\bar{s}' \in \bar{S}}\left[\bar{T}(\bar{s}'|\bar{s},a) - \sum_{s' \in \bar{s}'} T(s'|s,a)\right]\bar{V}^{*,n-1}(\bar{s}')| \tag{79}$$

$$\leq \eta_R + (n-1)\eta_R + \frac{(n-1-1)(n-1)}{2}\eta_T|\bar{S}|R_{\max} + \eta_T(n-1)|\bar{S}|R_{\max} \tag{80}$$

$$= n\eta_R + \frac{(n-1)n}{2}\eta_T|\bar{S}|R_{\max}. \tag{81}$$

For the step from equation 77 to equation 78, we subtract and add $\sum_{\bar{s}' \in \bar{S}} \bar{V}^{*,n-1}(\bar{s}') \sum_{s' \in \bar{s}'} T(s'|s,a)$, and from equation 79 to equation 80, we use the inductive hypothesis and again the fact that $(n-1)R_{\max}$ is an upper bound on $\bar{V}^{*,n-1}(\bar{s}')$ since the maximum reward per timestep is $R_{\max}$.

Now, for the discounted infinite horizon. By induction, we will show that for $n \geq 1$

$$\forall \bar{s} \in \bar{S}, s \in \bar{s} : |V^{\bar{\pi},n}(s) - \bar{V}^{\bar{\pi},n}(s)| \leq \eta_R\gamma^{n-1} + \sum_{i=0}^{n-2}\gamma^i(\eta_R + \frac{\gamma\eta_T|\bar{S}|R_{\max}}{1-\gamma}). \tag{82}$$

For $n = 1$, we have

$$|V^{\bar{\pi},1}(s) - \bar{V}^{\bar{\pi},1}(s)| = |\max_{a \in A} R(s,a) - \max_{a \in A}\bar{R}(\bar{s},a)| \leq \max_{a \in A}|R(s,a) - \bar{R}(\bar{s},a)| \leq \eta_R. \tag{83}$$

Now assume that the induction hypothesis, equation 82, holds for $n-1$, then

$$|V^{*,n}(s) - \bar{V}^{*,n}(\bar{s})| = \max_{a \in A} |R(s,a) - \bar{R}(\bar{s},a) + \gamma \sum_{s' \in S} T(s'|s,a)V^{*,n-1}(s') - \gamma \sum_{\bar{s}' \in \bar{S}} \bar{T}(\bar{s}'|\bar{s},a)\bar{V}^{*,n-1}(\bar{s}')| \quad (84)$$

$$\leq \max_{a \in A} |R(s,a) - \bar{R}(\bar{s},a)| + \max_{a \in A} \gamma | \sum_{\bar{s}' \in \bar{S}} \sum_{s' \in \bar{s}'} T(s'|s,a)V^{*,n-1}(s') - \sum_{\bar{s}' \in \bar{S}} \bar{T}(\bar{s}'|\bar{s},a)\bar{V}^{*,n-1}(\bar{s}')| \quad (85)$$

$$\leq \eta_R + \max_{a \in A} \gamma | \sum_{\bar{s}' \in \bar{S}} \sum_{s' \in \bar{s}'} T(s'|s,a)V^{*,n-1}(s') - \sum_{\bar{s}' \in \bar{S}} \bar{V}^{*,n-1}(\bar{s}') \sum_{s' \in \bar{s}'} T(s'|s,a)|$$

$$+ \max_{a \in A} \gamma | \sum_{\bar{s}' \in \bar{S}} \bar{V}^{*,n-1}(\bar{s}') \sum_{s' \in \bar{s}'} T(s'|s,a) - \sum_{\bar{s}' \in \bar{S}} \bar{T}(\bar{s}'|\bar{s},a)\bar{V}^{*,n-1}(\bar{s}')| \quad (86)$$

$$\leq \eta_R + \max_{a \in A} \gamma | \sum_{\bar{s}' \in \bar{S}} \sum_{s' \in \bar{s}'} T(s'|s,a)\big[V^{*,n-1}(s') - \bar{V}^{*,n-1}(\bar{s}')\big]| + \max_{a \in A} \gamma | \sum_{\bar{s}' \in \bar{S}} \big[\bar{T}(\bar{s}'|\bar{s},a) - \sum_{s' \in \bar{s}'} T(s'|s,a)\big]\bar{V}^{*,n-1}(\bar{s}')|$$

$$(87)$$

$$\leq \eta_R + \gamma\Big(\eta_R\gamma^{n-2} + \sum_{i=0}^{n-3} \gamma^i(\eta_R + \frac{\gamma\eta_T|\bar{S}|R_{\max}}{1-\gamma})\Big) + \gamma\eta_T|\bar{S}|\frac{R_{\max}}{1-\gamma} \quad (88)$$

$$= \eta_R + \eta_R\gamma^{n-1} + \sum_{i=1}^{n-2} \gamma^i(\eta_R + \frac{\gamma\eta_T|\bar{S}|R_{\max}}{1-\gamma}) + \gamma\eta_T|\bar{S}|\frac{R_{\max}}{1-\gamma} \quad (89)$$

$$= \eta_R\gamma^{n-1} + \sum_{i=0}^{n-2} \gamma^i(\eta_R + \frac{\gamma\eta_T|\bar{S}|R_{\max}}{1-\gamma}). \quad (90)$$

For the step from equation 85 to equation 86, we subtract and add $\sum_{\bar{s}' \in \bar{S}} \bar{V}^{\bar{\pi},n-1}(\bar{s}') \sum_{s' \in \bar{s}'} T(s'|s, \bar{\pi}(\bar{s}))$, and from equation 87 to equation 88, we use the inductive hypothesis and the fact that $\frac{R_{\max}}{1-\gamma}$ is an upper bound on $\bar{V}^{*,n-1}(\bar{s}')$.

Finally, taking the limit for $n \to \infty$, we get:

$$|V^*(s) - \bar{V}^*(s)| \leq \eta_R \times 0 + \frac{1}{1-\gamma}(\eta_R + \frac{\gamma\eta_T|\bar{S}|R_{\max}}{1-\gamma})$$

$$= \frac{\eta_R}{1-\gamma} + \frac{\gamma\eta_T|\bar{S}|R_{\max}}{(1-\gamma)^2}.$$

$\square$

*Proof of Theorem 1.* We can now proof Theorem 1, by using the triangle inequality and the results of Lemmas 6 and 7. For the undiscounted finite horizon:

$$|V^{*,h}(s) - V^{\bar{\pi}^*,h}(s)| \leq |V^{*,h}(s) - \bar{V}^{*,h}(s)| + |\bar{V}^{*,h}(s) - V^{\bar{\pi}^*,h}(s)|$$

$$= |V^{*,h}(s) - \bar{V}^{*,h}(s)| + |\bar{V}^{\bar{\pi}^*,h}(s) - V^{\bar{\pi}^*,h}(s)|$$

$$\leq h\eta_R + \frac{(h-1)h}{2}\eta_T|\bar{S}|R_{\max} + h\eta_R + \frac{(h-1)h}{2}\eta_T|\bar{S}|R_{\max}$$

$$= 2h\eta_R + (h-1)h\eta_T|\bar{S}|R_{\max}.$$

For the discounted infinite horizon:

$$|V^*(s) - V^{\bar{\pi}^*}(s)| \leq |V^*(s) - \bar{V}^*(s)| + |\bar{V}^*(s) - V^{\bar{\pi}^*}(s)|$$

$$= |V^*(s) - \bar{V}^*(s)| + |\bar{V}^{\bar{\pi}^*}(s) - V^{\bar{\pi}^*}(s)|$$

$$\leq \frac{\eta_R}{1-\gamma} + \frac{\gamma\eta_T|\bar{S}|R_{\max}}{(1-\gamma)^2} + \frac{\eta_R}{1-\gamma} + \frac{\gamma\eta_T|\bar{S}|R_{\max}}{(1-\gamma)^2}$$

$$= \frac{2\eta_R}{1-\gamma} + \frac{2\gamma\eta_T|\bar{S}|R_{\max}}{(1-\gamma)^2}.$$

$\square$

### A.3.1 Value Difference for Similar MDPs

Finally, we give a simulation lemma for two MDPs on the same state-action space.

**Lemma 8.** *Let $M$ and $M'$ be two MDPs on the same state-action space, with*

$$\forall s, a \in S \times A : \ |R_M(s,a) - R_{M'}(s,a)| \leq \eta_R, \tag{91}$$

$$\forall s, a, s' \in S \times A \times S : \ |T_M(s'|s,a) - T_{M'}(s'|s,a)| \leq \eta_T. \tag{92}$$

*Then, for every policy $\pi$ and for every state $s \in S$ we have:*

$$|V_M^{\pi,n}(s) - V_{M'}^{\pi,n}(s)| \leq n\eta_R + \frac{(n-1)n}{2}\eta_T|S|R_{max}. \tag{93}$$

*Proof.* By induction, we will show that for $n \geq 1$

$$\forall s \in S : \ |V_M^{\pi,n}(s) - V_{M'}^{\pi,n}(s)| \leq n\eta_R + \frac{(n-1)n}{2}\eta_T|S|R_{\max}. \tag{94}$$

For $n = 1$, we have

$$|V_M^{\pi,1}(s) - V_{M'}^{\pi,1}(s)| = |R_M(s,\pi(s)) - R_{M'}(s,\pi(s))| \leq \eta_R. \tag{95}$$

Now assume that the induction hypothesis, equation 94, holds for $n-1$, then

$$|V_M^{\pi,n}(s) - V_{M'}^{\pi,n}(s)| = |R_M(s,\pi(\bar{s})) - R_{M'}(s,\pi(s)) + \sum_{s' \in S} T_M(s'|s,\pi(s))V_M^{\pi,n-1}(s') - \sum_{s' \in S} T_{M'}(s'|s,\pi(s))V_{M'}^{\pi,n-1}(s')| \tag{96}$$

$$\leq |R_M(s,\pi(s)) - R_{M'}(s,\pi(s))| + |\sum_{s' \in S} T_M(s'|s,\pi(s))V_M^{\pi,n-1}(s') - \sum_{s' \in S} T_{M'}(s'|s,\pi(s))V_{M'}^{\pi,n-1}(s')| \tag{97}$$

$$\leq \eta_R + |\sum_{s' \in S} T_M(s'|s,\pi(s))V_M^{\pi,n-1}(s') - \sum_{s' \in S} T_M(s'|s,\pi(s))V_{M'}^{\pi,n-1}(s')|$$

$$+ |\sum_{s' \in S} T_M(s'|s,\pi(s))V_{M'}^{\pi,n-1}(s') - \sum_{s' \in S} T_{M'}(s'|s,\pi(s))V_{M'}^{\pi,n-1}(s')| \tag{98}$$

$$\leq \eta_R + |\sum_{s' \in S} T_M(s'|s,\pi(s))\big[V_M^{\pi,n-1}(s') - V_{M'}^{\pi,n-1}(s')\big]| + |\sum_{s' \in S}\big[T_M(s'|s,\pi(s)) - T_{M'}(s'|s,\pi(s))\big]V_{M'}^{\pi,n-1}(s')| \tag{99}$$

$$\leq \eta_R + (n-1)\eta_R + \frac{(n-1-1)(n-1)}{2}\eta_T|S|R_{\max} + \eta_T(n-1)|S|R_{\max} \tag{100}$$

$$= n\eta_R + \frac{(n-2)(n-1)}{2}\eta_T|S|R_{\max} + \eta_T(n-1)|S|R_{\max} \tag{101}$$

$$= n\eta_R + \frac{(n-1)n}{2}\eta_T|S|R_{\max}. \tag{102}$$

For the step from equation 97 to equation 98, we add and subtract $\sum_{\bar{s}' \in S} T_M(s'|s,\pi(s))V_{M'}^{\pi,n-1}(s')$, and from equation 99 to equation 100, we use the inductive hypothesis and the fact that $(n-1)R_{\max}$ is an upper bound on $V_{M'}^{\pi,n-1}(s')$ since the maximum reward per timestep is $R_{\max}$. $\qquad\square$

This shows that the values under any policy are similar for similar MDPs.

## B  L1 Inequality for Independent but Not Identically Distributed Variables

We show that we can adapt the proof of Weissman et al. (2003) for independent, but not identically distributed, samples to obtain the following result:

**Lemma 2.** *Let $X_{\bar{s},a} = s_1, \cdots, s_m$ be a sequence of states $s \in \bar{s}$ and let $\bar{\boldsymbol{Y}}_{\bar{s},a} = \bar{Y}^{(1)}, \bar{Y}^{(2)}, \cdots, \bar{Y}^{(m)}$ be independent random variables distributed according to $\Pr(\cdot|s_1,a), \cdots, \Pr(\cdot|s_m,a)$. Then, $\forall \epsilon > 0$,*

$$\Pr(||\bar{T}_Y(\cdot|\bar{s},a) - \bar{T}_{\omega_X}(\cdot|\bar{s},a)||_1 \geq \epsilon) \leq (2^{|\bar{S}|} - 2)e^{-\frac{1}{2}m\epsilon^2}. \tag{103}$$

*Proof of Lemma 2.* The proof mostly follows the steps by Weissman et al. (2003).

To shorten the notation, we define $P_Y \triangleq \bar{T}_Y(\cdot|\bar{s}, a)$ and $P_{\omega_X} \triangleq \bar{T}_{\omega_X}(\cdot|\bar{s}, a)$.

We will make use of the following result (Proposition 4.2 in (Levin & Peres, 2017)), that for any distribution $Q$ on $\bar{S}$

$$||Q - P_{\omega_X}||_1 = 2 \max_{\bar{\mathcal{S}} \subseteq \bar{S}} (Q(\bar{\mathcal{S}}) - P_{\omega_X}(\bar{\mathcal{S}})),$$

where $\bar{\mathcal{S}}$ is a subset of $\bar{S}$, and $P_{\omega_X}(\bar{\mathcal{S}}) = \sum_{\bar{s}' \in \bar{\mathcal{S}}} P_{\omega_X}(\bar{s}')$. Thus, we have that

$$||P_Y - P_{\omega_X}||_1 = 2 \max_{\bar{\mathcal{S}} \subseteq \bar{S}} (P_Y(\bar{\mathcal{S}}) - P_{\omega_X}(\bar{\mathcal{S}})). \tag{104}$$

Using this, we can write

$$\Pr(||P_Y - P_{\omega_X}||_1 \geq \epsilon) = \Pr\left[2 \max_{\bar{\mathcal{S}} \subseteq \bar{S}} \left[P_Y(\bar{\mathcal{S}}) - P_{\omega_X}(\bar{\mathcal{S}})\right] \geq \epsilon\right] \tag{105}$$

$$= \Pr\left[\max_{\bar{\mathcal{S}} \subseteq \bar{S}} \left[P_Y(\bar{\mathcal{S}}) - P_{\omega_X}(\bar{\mathcal{S}})\right] \geq \frac{\epsilon}{2}\right] \tag{106}$$

$$= \Pr\left[\cup_{\bar{\mathcal{S}} \subseteq \bar{S}} \left[P_Y(\bar{\mathcal{S}}) - P_{\omega_X}(\bar{\mathcal{S}}) \geq \frac{\epsilon}{2}\right]\right] \tag{107}$$

$$\leq \sum_{\bar{\mathcal{S}} \subseteq \bar{S}} \Pr\left[P_Y(\bar{\mathcal{S}}) - P_{\omega_X}(\bar{\mathcal{S}}) \geq \frac{\epsilon}{2}\right], \tag{108}$$

where the last step follows from the union bound (Lemma 5).

Assuming $\epsilon > 0$, we have that $\Pr(P_Y(\bar{\mathcal{S}}) - P_{\omega_X}(\bar{\mathcal{S}}) \geq \frac{\epsilon}{2}) = 0$ when $\bar{\mathcal{S}} = \bar{S}$ or $\bar{\mathcal{S}} = \emptyset$. For every other subset $\bar{\mathcal{S}}$, we can define a random binary variable that is 1 when $Y^{(i)} \in \bar{\mathcal{S}}$ and 0 otherwise. Here $P_{\omega_X}(\bar{\mathcal{S}})$ acts as $\mu$ (equation 39) from Lemma 4 and $P_Y(\bar{\mathcal{S}})$ as $\bar{Z}$ (equation 38). Thus, by applying Lemma 4 to this random variable, we have

$$\Pr(P_Y(\bar{\mathcal{S}}) - P_{\omega_X}(\bar{\mathcal{S}}) \geq \frac{\epsilon}{2}) \leq e^{-2m\frac{\epsilon}{2}^2} = e^{-\frac{1}{2}m\epsilon^2}. \tag{109}$$

Then it follows that

$$\Pr(||P_Y - P_{\omega_X}||_1 \geq \epsilon) \leq \sum_{\bar{\mathcal{S}} \subseteq \bar{S}} \Pr(P_Y(\bar{\mathcal{S}}) - P_{\omega_X}(\bar{\mathcal{S}}) \geq \frac{\epsilon}{2}) \tag{110}$$

$$\leq \sum_{\bar{\mathcal{S}} \subset \bar{S}: \bar{\mathcal{S}} \neq \bar{S}, \emptyset} \Pr(P_Y(\bar{\mathcal{S}}) - P_{\omega_X}(\bar{\mathcal{S}}) \geq \frac{\epsilon}{2}) \tag{111}$$

$$\leq (2^{|\bar{S}|} - 2)e^{-\frac{1}{2}m\epsilon^2}, \tag{112}$$

where $\bar{\mathcal{S}} \subset \bar{S}: \bar{\mathcal{S}} \neq \bar{S}, \emptyset$ denotes that the empty set $\emptyset$ and the complete set $\bar{S}$ are excluded. $\quad\square$

## C   Proof of Main Result

Here we show how we can use a concentration inequality for martingales to learn an accurate transition model in RLAO. Specifically, the following result shows that, with a high probability, the empirical abstract transition function $\bar{T}_Y$ will be close to the abstract transition function $\bar{T}_{\omega_X}$. In the proof, which follows the general approach of Ortner et al. (2020), we define a suitable martingale difference sequence for the abstract transition function and use this to obtain the following result for learning a transition function in RLAO:

**Theorem 2** (Abstract L1 inequality). *If an agent has access to a state abstraction function $\phi$ and uses this to collect data for any abstract state-action pair $(\bar{s}, a)$ by acting in an MDP $M$ according to a policy $\bar{\pi}$, we have that the following holds with a probability of at least $1 - \delta$ for a fixed value of $N(\bar{s}, a)$:*

$$||\bar{T}_Y(\cdot|\bar{s}, a) - \bar{T}_{\omega_X}(\cdot|\bar{s}, a)||_1 \leq \epsilon, \tag{113}$$

*where $\delta = 2^{|\bar{S}|}e^{-\frac{1}{8}N(\bar{s}, a)\epsilon^2}$.*

*Proof of Theorem 2.* We first define an abstract transition function based on $X_{\bar{s},a}$ as

$$\forall (\bar{s}, a), \bar{s}' : \bar{T}_{\omega_X}(\bar{s}'|\bar{s}, a) \triangleq \frac{1}{N(\bar{s}, a)} \sum_{i=1}^{N(\bar{s},a)} T(\bar{s}'|X_{\bar{s},a}^{(i)}, a), \tag{114}$$

where $T(\bar{s}'|X_{\bar{s},a}^{(i)}, a) \triangleq \sum_{s' \in \bar{s}'} T(s'|X_{\bar{s},a}^{(i)}, a)$. We write $\bar{T}_{\omega_X}$ because this definition is equivalent to using a weighting function as in equation 19:

$$\forall (\bar{s}, a), \bar{s}' : \bar{T}_{\omega_X}(\bar{s}'|\bar{s}, a) \triangleq \sum_{s \in \bar{s}} \omega_X(s, a) \sum_{s' \in \bar{s}'} T(s'|s, a) \tag{Eq. 19} \tag{115}$$

$$= \sum_{s \in \bar{s}} \frac{1}{N(\bar{s}, a)} \sum_{i=1}^{N(\bar{s},a)} \mathbb{1}\{X_{\bar{s},a}^{(i)} = s\} \sum_{s' \in \bar{s}'} T(s'|s, a) \tag{116}$$

$$= \frac{1}{N(\bar{s}, a)} \sum_{i=1}^{N(\bar{s},a)} \sum_{s \in \bar{s}} \mathbb{1}\{X_{\bar{s},a}^{(i)} = s\} \sum_{s' \in \bar{s}'} T(s'|s, a) \tag{117}$$

$$= \frac{1}{N(\bar{s}, a)} \sum_{i=1}^{N(\bar{s},a)} \sum_{s' \in \bar{s}'} T(s'|X_{\bar{s},a}^{(i)}, a) \tag{118}$$

$$= \frac{1}{N(\bar{s}, a)} \sum_{i=1}^{N(\bar{s},a)} T(\bar{s}'|X_{\bar{s},a}^{(i)}, a). \tag{Eq. 114} \tag{119}$$

Now we use $\boldsymbol{z}$ to denote a vector of size $|\bar{S}|$ with entries $\pm 1$, and we write $z(\bar{s})$ for the entry in $\boldsymbol{z}$ with index $\bar{s}$. Then we have

$$||\bar{T}_Y(\cdot|\bar{s}, a) - \bar{T}_{\omega_X}(\cdot|\bar{s}, a)||_1 = \sum_{\bar{s}'} |\bar{T}_Y(\bar{s}'|\bar{s}, a) - \bar{T}_{\omega_X}(\bar{s}'|\bar{s}, a)| \tag{120}$$

$$= \max_{\boldsymbol{z} \in \{-1,1\}^{\bar{S}}} \sum_{\bar{s}'} \left( \bar{T}_Y(\bar{s}'|\bar{s}, a) - \bar{T}_{\omega_X}(\bar{s}'|\bar{s}, a) \right) z(\bar{s}') \tag{121}$$

$$= \max_{\boldsymbol{z} \in \{-1,1\}^{\bar{S}}} \sum_{\bar{s}'} \left( \frac{1}{N(\bar{s}, a)} \sum_{i=1}^{N(\bar{s},a)} \mathbb{1}\{\bar{Y}_{\bar{s},a}^{(i)} = \bar{s}'\} - \frac{1}{N(\bar{s}, a)} \sum_{i=1}^{N(\bar{s},a)} T(\bar{s}'|X_{\bar{s},a}^{(i)}, a) \right) z(\bar{s}') \tag{122}$$

$$= \max_{\boldsymbol{z} \in \{-1,1\}^{\bar{S}}} \sum_{\bar{s}'} \frac{1}{N(\bar{s}, a)} \sum_{i=1}^{N(\bar{s},a)} \mathbb{1}\{\bar{Y}_{\bar{s},a}^{(i)} = \bar{s}'\} z(\bar{s}') - \sum_{\bar{s}'} \frac{1}{N(\bar{s}, a)} \sum_{i=1}^{N(\bar{s},a)} T(\bar{s}'|X_{\bar{s},a}^{(i)}, a) z(\bar{s}') \tag{123}$$

$$= \max_{\boldsymbol{z} \in \{-1,1\}^{\bar{S}}} \frac{1}{N(\bar{s}, a)} \sum_{i=1}^{N(\bar{s},a)} z(\bar{Y}_{\bar{s},a}^{(i)}) - \frac{1}{N(\bar{s}, a)} \sum_{i=1}^{N(\bar{s},a)} \sum_{\bar{s}'} T(\bar{s}'|X_{\bar{s},a}^{(i)}, a) z(\bar{s}') \tag{124}$$

$$= \max_{\boldsymbol{z} \in \{-1,1\}^{\bar{S}}} \frac{1}{N(\bar{s}, a)} \sum_{i=1}^{N(\bar{s},a)} \left( z(\bar{Y}_{\bar{s},a}^{(i)}) - \sum_{\bar{s}'} T(\bar{s}'|X_{\bar{s},a}^{(i)}, a) z(\bar{s}') \right) \tag{125}$$

$$= \max_{\boldsymbol{z} \in \{-1,1\}^{\bar{S}}} \frac{1}{N(\bar{s}, a)} \sum_{i=1}^{N(\bar{s},a)} Z_{\tau_i}(\boldsymbol{z}, X_{\bar{s},a}^{(i)}, a, \bar{Y}_{\bar{s},a}^{(i)}), \tag{126}$$

where we set

$$Z_{\tau_i}(\boldsymbol{z}, X_{\bar{s},a}^{(i)}, a_{\tau_i}, \bar{Y}_{\bar{s},a}^{(i)}) \triangleq z(\bar{Y}_{\bar{s},a}^{(i)}) - \sum_{\bar{s}'} T(\bar{s}'|X_{\bar{s},a}^{(i)}, a_{\tau_i}) z(\bar{s}').$$

To show that $\sum_i^{N(\bar{s},a)} Z_{\tau_i}(\boldsymbol{z}, X_{\bar{s},a}^{(i)}, a_{\tau_i}, \bar{Y}_{\bar{s},a}^{(i)})$ is a martingale difference sequence, we should follow Definition 3 and show that $\forall i : E[Z_{\tau_i}(\boldsymbol{z}, X_{\bar{s},a}^{(i)}, a_{\tau_i}, \bar{Y}_{\bar{s},a}^{(i)})|Z_{\tau_1}, Z_{\tau_2}, \cdots, Z_{\tau_{i-1}}] = 0$ and $|Z_i| < \infty$. For the second part, we

have that $\forall i : |Z_{\tau_i}(\boldsymbol{z}, X_{\bar{s},a}^{(i)}, a_{\tau_i}, \bar{Y}_{\bar{s},a}^{(i)})| \leq 2$, since $|z(\bar{Y}_{\bar{s},a}^{(i)})| \leq 1$ and $|\sum_{\bar{s}'} T(\bar{s}'|X_{\bar{s},a}^{(i)}, a_{\tau_i})z(\bar{s}')| \leq 1$. For the first part, we use the following Lemma, the proof of which follows after the current proof.

**Lemma 9.** *Let $\pi$ be a policy, and suppose the sequence $s_1, a_1 \cdots, s_{t-1}, a_{t-1}, s_t$ is to be generated by $\pi$. If $1 \leq \tau_1 < \tau_2 < \cdots < \tau_{i-1} < \tau_i \leq t$, then $E[Z_{\tau_i}(\boldsymbol{z}, X_{\bar{s},a}^{(i)}, a_{\tau_i}, \bar{Y}_{\bar{s},a}^{(i)})|Z_{\tau_1}, Z_{\tau_2}, \cdots, Z_{\tau_{i-1}}] = 0.$*

Lemma 9 shows that $\sum_i^{N(\bar{s},a)} Z_{\tau_i}(\boldsymbol{z}, X_{\bar{s},a}^{(i)}, a_{\tau_i}, \bar{Y}_{\bar{s},a}^{(i)})$ is a martingale difference sequence with $\forall i :$ $|Z_{\tau_i}(\boldsymbol{z}, X_{\bar{s},a}^{(i)}, a_{\tau_i}, \bar{Y}_{\bar{s},a}^{(i)})| \leq 2$ for any fixed $\boldsymbol{z}$ and fixed $N(\bar{s}, a) = n$ so that by the Azuma-Hoeffding inequality (Lemma 3):

$$\Pr\left(\sum_{i=1}^{N(\bar{s},a)} Z_{\tau_i} > \epsilon\right) \leq e^{-\frac{\epsilon^2}{8N(\bar{s},a)}}. \tag{127}$$

Similarly, $\sum_i^{N(\bar{s},a)} \frac{1}{N(\bar{s},a)} Z_{\tau_i}(\boldsymbol{z}, X_{\bar{s},a}^{(i)}, a_{\tau_i}, \bar{Y}_{\bar{s},a}^{(i)})$ is a martingale difference sequence with $\forall i :$ $|\frac{1}{N(\bar{s},a)} Z_{\tau_i}(\boldsymbol{z}, X_{\bar{s},a}^{(i)}, a_{\tau_i}, \bar{Y}_{\bar{s},a}^{(i)})| \leq \frac{2}{N(\bar{s},a)}$ for any fixed $\boldsymbol{z}$ and $N(\bar{s}, a) = n$ so that, by the Azuma-Hoeffding inequality (Lemma 3), the following holds:

$$\Pr\left(\frac{1}{N(\bar{s},a)} \sum_{i=1}^{N(\bar{s},a)} Z_{\tau_i} > \epsilon\right) \leq e^{-\frac{\epsilon^2}{2\frac{4}{N(\bar{s},a)^2}N(\bar{s},a)}} \tag{128}$$

$$= e^{-\frac{\epsilon^2}{\frac{8}{N(\bar{s},a)}}} \tag{129}$$

$$= e^{-\frac{1}{8}N(\bar{s},a)\epsilon^2}. \tag{130}$$

From equation 120 and equation 126, we then obtain

$$\Pr(||\bar{T}_Y(\cdot|\bar{s},a) - \bar{T}_{\omega_X}(\cdot|\bar{s},a)||_1 > \epsilon) = \Pr\left(\max_{\boldsymbol{z} \in \{-1,1\}^S} \frac{1}{N(\bar{s},a)} \sum_{i=1}^{N(\bar{s},a)} Z_{\tau_i} > \epsilon\right). \tag{131}$$

A union bound (Lemma 5) over all $2^{|\bar{S}|}$ vectors $\boldsymbol{z}$ for a fixed value of $N(s, a)$ shows

$$\Pr\left(\max_{\boldsymbol{z} \in \{-1,1\}^S} \frac{1}{N(\bar{s},a)} \sum_{i=1}^{N(\bar{s},a)} Z_{\tau_i} > \epsilon\right) \leq \sum_{\boldsymbol{z} \in \{-1,1\}^S} \Pr\left(\frac{1}{N(\bar{s},a)} \sum_{i=1}^{N(\bar{s},a)} Z_{\tau_i} > \epsilon\right). \tag{132}$$

So, using equation 130, we have that the following holds with probability $1 - 2^{|\bar{S}|}e^{-\frac{1}{8}N(\bar{s},a)\epsilon^2}$:

$$||\bar{T}_Y(\cdot|\bar{s},a) - \bar{T}_{\omega_X}(\cdot|\bar{s},a)||_1 \leq \epsilon. \tag{133}$$

$\square$

Now we give the proof of Lemma 9:

*Proof of Lemma 9.* We follow the general structure of the proof of Lemma 8 by Strehl & Littman (2008). We have

$$E[Z_{\tau_i}(\boldsymbol{z}, X_{\bar{s},a}^{(i)}, a_{\tau_i}, \bar{Y}_{\bar{s},a}^{(i)})] = \sum_{c_{\tau_i+1}} \Pr(c_{\tau_i+1})Z_{\tau_i}(\boldsymbol{z}, X_{\bar{s},a}^{(i)}, a_{\tau_i}, \bar{Y}_{\bar{s},a}^{(i)}) \tag{134}$$

$$= \sum_{c_{\tau_i}} \Pr(c_{\tau_i}) \sum_{\bar{Y}_{\bar{s},a}^{(i)}} \Pr(\bar{Y}_{\bar{s},a}^{(i)}|c_{\tau_i}, a_{\tau_i})Z_{\tau_i}(\boldsymbol{z}, X_{\bar{s},a}^{(i)}, a_{\tau_i}, \bar{Y}_{\bar{s},a}^{(i)}) \tag{135}$$

$$= \sum_{c_{\tau_i}} \Pr(c_{\tau_i}) \sum_{\bar{Y}_{\bar{s},a}^{(i)}} \Pr(\bar{Y}_{\bar{s},a}^{(i)}|X_{\bar{s},a}^{(i)}, a_{\tau_i})Z_{\tau_i}(\boldsymbol{z}, X_{\bar{s},a}^{(i)}, a_{\tau_i}, \bar{Y}_{\bar{s},a}^{(i)}). \tag{136}$$

The sum $\sum_{c_{\tau_i+1}}$ is over all possible sequences $c_{\tau_i+1}$ that end in a state $\bar{s}_{\tau_i+1}$, resulting from $\tau_i$ actions chosen by an agent following policy $\pi$. Conditioning on the sequence of random variables $Z_{\tau_1}, Z_{\tau_2}, \cdots, Z_{\tau_{i-1}}$ can make some sequences $c_{\tau_i}$ more likely and others less likely, that is

$$E[Z_{\tau_i}(\boldsymbol{z}, X_{\bar{s},a}^{(i)}, a_{\tau_i}, \bar{Y}_{\bar{s},a}^{(i)})|Z_{\tau_1}, Z_{\tau_2}, \cdots, Z_{\tau_{i-1}}] \tag{137}$$

$$= \sum_{c_{\tau_i}} \Pr(c_{\tau_i}|Z_{\tau_1}, Z_{\tau_2}, \cdots, Z_{\tau_{i-1}}) \sum_{\bar{Y}_{\bar{s},a}^{(i)}} \Pr(\bar{Y}_{\bar{s},a}^{(i)}|X_{\bar{s},a}^{(i)}, a_{\tau_i}) Z_{\tau_i}(\boldsymbol{z}, X_{\bar{s},a}^{(i)}, a_{\tau_i}, \bar{Y}_{\bar{s},a}^{(i)}). \tag{138}$$

Significantly, since $P(\bar{Y}_{\bar{s},a}^{(i)}|\bar{s}_{\tau_i}, a_{\tau_i}, Z_{\tau_1}, \cdots, Z_{\tau_{i-1}}) = P(\bar{Y}_{\bar{s},a}^{(i)}|\bar{s}_{\tau_i}, a_{\tau_i})$, fixed values of $Z_{\tau_1}, Z_{\tau_2}, \cdots, Z_{\tau_{i-1}}$ do not influence the innermost sum of equation 138. For this innermost sum, we have

$$\sum_{\bar{Y}_{\bar{s},a}^{(i)}} \Pr(\bar{Y}_{\bar{s},a}^{(i)}|X_{\bar{s},a}^{(i)}, a_{\tau_i}) Z_{\tau_i}(\boldsymbol{z}, X_{\bar{s},a}^{(i)}, a_{\tau_i}, \bar{Y}_{\bar{s},a}^{(i)}) \tag{139}$$

$$= \sum_{\bar{Y}_{\bar{s},a}^{(i)}} \Pr(\bar{Y}_{\bar{s},a}^{(i)}|X_{\bar{s},a}^{(i)}, a_{\tau_i}) \left[ z(\bar{Y}_{\bar{s},a}^{(i)}) - \sum_{\bar{s}'} T(\bar{s}'|X_{\bar{s},a}^{(i)}, a_{\tau_i}) z(\bar{s}') \right] \tag{140}$$

$$= \sum_{\bar{Y}_{\bar{s},a}^{(i)}} \Pr(\bar{Y}_{\bar{s},a}^{(i)}|X_{\bar{s},a}^{(i)}, a_{\tau_i}) z(\bar{Y}_{\bar{s},a}^{(i)}) - \sum_{\bar{Y}_{\bar{s},a}^{(i)}} \Pr(\bar{Y}_{\bar{s},a}^{(i)}|X_{\bar{s},a}^{(i)}, a_{\tau_i}) \sum_{\bar{s}'} T(\bar{s}'|X_{\bar{s},a}^{(i)}, a_{\tau_i}) z(\bar{s}') \tag{141}$$

$$= \sum_{\bar{Y}_{\bar{s},a}^{(i)}} \Pr(\bar{Y}_{\bar{s},a}^{(i)}|X_{\bar{s},a}^{(i)}, a_{\tau_i}) z(\bar{Y}_{\bar{s},a}^{(i)}) - \sum_{\bar{s}'} T(\bar{s}'|X_{\bar{s},a}^{(i)}, a_{\tau_i}) z(\bar{s}') \tag{142}$$

$$= 0. \tag{143}$$

So we conclude

$$E[Z_{\tau_i}(\boldsymbol{z}, X_{\bar{s},a}^{(i)}, a_{\tau_i}, \bar{Y}_{\bar{s},a}^{(i)})|Z_{\tau_1}, Z_{\tau_2}, \cdots, Z_{\tau_{i-1}}] \tag{144}$$

$$= \sum_{c_{\tau_i}} \Pr(c_{\tau_i}|Z_{\tau_1}, Z_{\tau_2}, \cdots, Z_{\tau_{i-1}}) \sum_{\bar{s}_{\tau_i+1}} \Pr(\bar{s}_{\tau_i+1}|X_{\bar{s},a}^{(i)}, a_{\tau_i}) Z_{\tau_i}(\boldsymbol{z}, X_{\bar{s},a}^{(i)}, a_{\tau_i}, \bar{Y}_{\bar{s},a}^{(i)}) \tag{145}$$

$$= \sum_{c_{\tau_i}} \Pr(c_{\tau_i}|Z_{\tau_1}, Z_{\tau_2}, \cdots, Z_{\tau_{i-1}}) \times 0 \tag{146}$$

$$= 0. \tag{147}$$

$\square$

# D  R-MAX From Abstracted Observations

Here we use the result of Theorem 2 to show that we can provide efficient learning guarantees for R-MAX (Brafman & Tennenholtz, 2002) in RLAO. In Appendix D.1, we use Theorem 2 and the value bounds in Appendix A.3 to establish two supporting Lemmas. Then, in Appendix D.2, we adapt one lemma and the guarantees of R-MAX to RLAO.

## D.1  Supporting Lemmas

We can use Theorem 2 to determine the number of samples required to guarantee that the distance $||\bar{T}_Y(\cdot|\bar{s}, a) - \bar{T}_{\omega_X}(\cdot|\bar{s}, a)||_1$ will be smaller than $\epsilon$ with probability $1 - \delta$:

**Lemma 10.** *For inputs $\kappa$ and $\epsilon$ ($0 < \kappa < 1, 0 < \epsilon < 2$), the following holds for a number of samples $m \geq \frac{2[\ln(2^{|\bar{S}|}-2)-\ln(\kappa)]}{\epsilon^2}$:*

$$\Pr(||\bar{T}_Y(\cdot|\bar{s}, a) - \bar{T}_{\omega_X}(\cdot|\bar{s}, a)||_1 \geq \epsilon) \leq \kappa. \tag{148}$$

*Proof.* To shorten the notation, we use the definitions $P_Y \triangleq \bar{T}_Y(\cdot|\bar{s}, a)$ and $P_{\omega_X} \triangleq \bar{T}_{\omega_X}(\cdot|\bar{s}, a)$. It follows from Theorem 2 that

$$\Pr(||P_Y - P_{\omega_X}||_1 \geq \epsilon) \leq 2^{|\bar{S}|} e^{-\frac{1}{8} m \epsilon^2}. \tag{149}$$

We need to select $m$ such that $\kappa \geq 2^{|\bar{S}|} e^{-\frac{1}{8} m \epsilon^2}$:

$$\kappa \geq 2^{|\bar{S}|} e^{-\frac{1}{8} m \epsilon^2} \tag{150}$$

$$\frac{\kappa}{2^{|\bar{S}|}} \geq e^{-\frac{1}{8} m \epsilon^2} \tag{151}$$

$$\ln(\kappa) - \ln(2^{|\bar{S}|}) \geq -\frac{m \epsilon^2}{8} \tag{152}$$

$$\frac{m \epsilon^2}{8} \geq \ln(2^{|\bar{S}|}) - \ln(\kappa) \tag{153}$$

$$m \geq \frac{8[\ln(2^{|\bar{S}|}) - \ln(\kappa)]}{\epsilon^2}. \tag{154}$$

Thus if $m \geq \frac{8[\ln(2^{|\bar{S}|}) - \ln(\kappa)]}{\epsilon^2}$, we have

$$\Pr(||P_Y - P_{\omega_X}||_1 \geq \epsilon) \leq \kappa. \qquad \square$$

We want to give results for an empirical abstract model $\hat{\bar{M}}$ in the abstract space from $\phi$, whose transition probabilities and rewards are within $\eta_T$ and $\eta_R$, respectively, from those of an abstract MDP $\bar{M}$. We use $V^{*,n}$ to denote the value in $M$ under the n-step optimal policy and $V^{\hat{\bar{\pi}}^*,n}$ to denote the value in $M$ under the n-step optimal policy $\hat{\bar{\pi}}^*$ for $\hat{\bar{M}}$. The following lemma shows that we can upper bound the loss in value when applying $\hat{\bar{\pi}}^*$ to $M$:

**Lemma 11.** *Let $M$ be an MDP, $\bar{M}$ an abstract MDP constructed using an approximate model similarity abstraction $\phi$, with $\eta_R$ and $\eta_T$, and $\hat{\bar{M}}$ an MDP in the abstract space from $\phi$ with*

$$|\bar{T}(\bar{s}'|\bar{s}, a) - \hat{\bar{T}}(\bar{s}'|\bar{s}, a)| \leq \epsilon, |\bar{R}(\bar{s}, a) - \hat{\bar{R}}(\bar{s}, a)| = 0. \tag{155}$$

*Then*

$$V^{*,n}(s) - V^{\hat{\bar{\pi}}^*,n}(s) \leq 2n\eta_R + (n-1)n(\eta_T + \epsilon)|\bar{S}|R_{max}. \tag{156}$$

*Proof.* Note that we assume that $|\bar{R}(\bar{s}, a) - \hat{\bar{R}}(\bar{s}, a)| = 0$ because we assume a deterministic reward. Then, we have

$$\forall \bar{s}, a \in \bar{S} \times A, s \in \bar{s}: |R(s, a) - \hat{\bar{R}}(\bar{s}, a)| \leq \eta_R, \tag{157}$$

$$\forall \bar{s}, a, \bar{s}' \in \bar{S} \times A \times \bar{S}, s \in \bar{s}: |\sum_{s' \in \bar{s}'} T(s'|s, a) - \hat{\bar{T}}(\bar{s}'|\bar{s}, a)| \leq \eta_T + \epsilon. \tag{158}$$

We use $\hat{\bar{V}}^{\hat{\bar{\pi}}^*,n}(\bar{s})$ to denote the n-step value under the n-step optimal policy $\hat{\bar{\pi}}^{*,n}$ for the empirical abstract MDP $\hat{\bar{M}}$. Then, by Theorem 1, we have $\forall s \in \bar{s}, \bar{s} \in \bar{S}$:

$$V^{*,n}(s) - V^{\hat{\bar{\pi}}^*,n}(s) = 2n\eta_R + (n-1)n(\eta_T + \epsilon)|\bar{S}|R_{\max}. \tag{159}$$

$$\square$$

## D.2 Proof of Theorem 4

First, we restate an Implicit Explore or Exploit Lemma that is used in the proof of R-MAX. We are interested in the event $A_M$, the event that we encounter an unknown state-action pair during an $n$-step trail in $M$. For two MDPs with different dynamics only in the unknown state-action pairs, the probability that we encounter an unknown state-action pair in an $n$-step trial is tiny if the difference in the n-step value between the two MDPs is slight. The proof follows the steps the proof of Lemma 3 from Strehl & Littman (2008).

**Lemma 12** (Implicit Explore or Exploit). *Let $M$ be an MDP. Let $L$ be the set of known abstract state-action pairs, and let $M_L$ be an MDP that is the same as $M$ on the known pairs $(\bar{s}, a) \in L$, but different on the unknown pairs $(\bar{s}, a) \notin L$. Let $s$ be some state, and $A_M$ the event that an unknown abstract state-action pair is encountered in a trial generated by starting from state $s_1$ and following $\pi$ for $n$ steps in $M$. Then,*

$$V_M^{\pi,n}(s_1) \geq V_{M_L}^{\pi,n}(s_1) - nR_{max} \Pr(A_M). \tag{160}$$

*Proof.* For a fixed path $p_t = s_1, a_1, r_1, \cdots, s_t, a_t, r_t$, we define $\Pr_M(p_t)$ as the probability that $p_t$ occurs when running policy $\pi$ in $M$ starting from state $s_1$. We let $L_t$ be the set of paths $p_t$ such that there is at least one unknown state $s_i$ in $p_t$ $((\phi(s_i), a) \notin L)$. We further let $r_M(t)$ be the reward received at time $t$ and $r_M(p_t, t)$ the reward at time $t$ in the path $p_t$. We have the following:

$$E\big[r_{M_L}(t)\big] - E\big[r_M(t)\big] = \sum_{p_t \in L_t} \big( \Pr_{M_L}(p_t) r_{M_L}(p_t, t) - \Pr_M(p_t) r_M(p_t, t) \big) \tag{161}$$

$$+ \sum_{p_t \notin L_t} \big( \Pr_{M_L}(p_t) r_{M_L}(p_t, t) - \Pr_M(p_t) r_M(p_t, t) \big) \tag{162}$$

$$= \sum_{p_t \notin L_t} \big( \Pr_{M_L}(p_t) r_{M_L}(p_t, t) - \Pr_M(p_t) r_M(p_t, t) \big) \tag{163}$$

$$\leq \sum_{p_t \notin L_t} \Pr_{M_L}(p_t) r_{M_L}(p_t, t) \leq R_{\max} \Pr(A_M). \tag{164}$$

Here $\sum_{p_t \in L_t} \big( \Pr_{M_L}(p_t) r_{M_L}(p_t, t) - \Pr_M(p_t) r_M(p_t, t) \big) = 0$ because, by definition, $M$ and $M_L$ behave identically on the known state-action pairs, and $\sum_{p_t \notin L_t} \Pr_{M_L}(p_t) r_{M_L}(p_t, t) \leq R_{\max} \Pr(A_M)$ is true because $r_{M_L}(p_t, t)$ is at most $R_{\max}$. Finally we can write

$$V_{M_L}^{\pi,n}(s_1) - V_M^{\pi,n}(s_1) = \sum_{t=0}^{n} (E\big[r_{M_L}(t)\big] - E\big[r_M(t)\big]) \tag{165}$$

$$\leq nR_{\max} \Pr(A_M). \tag{166}$$

Thus, $V_M^{\pi,n}(s_1) \geq V_{M_L}^{\pi,n}(s_1) - nR_{\max} \Pr(A_M)$. $\qquad\square$

Now we are ready to prove the theorem.

**Theorem 4.** *Given an MDP M, an approximate model similarity abstraction $\phi$, with $\eta_R$ and $\eta_T$, and inputs $|\bar{S}|, |A|, \epsilon, \delta, T_\epsilon$. With probability of at least $1 - \delta$ the R-MAX algorithm adapted to abstraction (Algorithm 1) will attain an expected return of $Opt(\prod_M(\epsilon, T_\epsilon)) - 3\frac{g(\eta_T, \eta_R)}{T_\epsilon} - 2\epsilon$ within a number of steps polynomial in $|\bar{S}|, |A|, \frac{1}{\epsilon}\frac{1}{\delta}, T_\epsilon$. Here $T_\epsilon$ is the $\epsilon$-return mixing time of the optimal policy, the policies for $M$ whose $\epsilon$-return mixing time is $T_\epsilon$ are denoted by $\prod_M(\epsilon, T_\epsilon)$, the optimal expected $T_\epsilon$-step undiscounted average return achievable by such policies are denoted by $Opt(\prod_M(\epsilon, T_\epsilon))$, and*

$$g(\eta_T, \eta_R) = T_\epsilon \eta_R + \frac{(T_\epsilon - 1)T_\epsilon}{2}\eta_T |\bar{S}| R_{max}.$$

*Proof of Theorem 4.* The proof uses elements of the Theorem from Brafman & Tennenholtz (2002). The proof follows the following steps:

1. We show that the expected average reward of the algorithm is at least as stated if the algorithm does not fail.

2. The probability of failing is at most $\delta$. We can decompose this probability into three elements.

   (a) Probability that the transition function estimates are not within the desired bounds.
   (b) The probability that we do not attain the number of required visits in polynomial time.
   (c) The probability that the actual return is lower than the expected return.

Now we first assume the algorithm does not fail. We define an abstract MDP $\bar{M}_{\omega_X}$ constructed from $\phi$ with $\eta_T$ and $\eta_R$. Similar to $M_L$, $\bar{M}_{\omega_X,L}$ is the same as $\bar{M}_{\omega_X}$ on the known abstract state-action pairs, but with a self-loop and the maximum reward ($R_{\max}$) on the unknown abstract state-action pairs, i.e., $\forall (\bar{s}, a) \notin L : \bar{T}_{\omega_X,L}(\bar{s}|\bar{s},a) = 1, \bar{R}_{\omega_X,L}(\bar{s},a) = R_{\max}$. We also define an empirical abstract MDP $\bar{M}_Y$, of which the transition probabilities $\bar{T}_Y(\bar{s}'|\bar{s},a)$ are within some $\epsilon_2$ (defined later) of those in $\bar{M}_{\omega_X}$ and with $\bar{R}_{\omega_X}(\bar{s},a) = \bar{R}_Y(\bar{s},a)$ because of the assumption that the rewards are deterministic. Then, $\bar{M}_{Y,L}$ is the abstract MDP that is the same as $\bar{M}_Y$ on the known abstract state-action pairs and the same as $\bar{M}_{\omega_X,L}$ on the unknown abstract state-action pairs. We denote the R-MAX policy with $\bar{\pi}$.

Let $A_M$ be the event that, following $\bar{\pi}$, we encounter an unknown abstract state-action pair $(\phi(s), a) \notin L$ in $T_\epsilon$ steps. From Lemma 12, we have that:

$$\forall s \in S : V_M^{\bar{\pi},n}(s) \geq V_{M_L}^{\bar{\pi},n}(s) - T_\epsilon R_{\max} \Pr(A_M). \tag{167}$$

Now suppose that $R_{\max} \Pr(A_M) < \epsilon_1$, for some $\epsilon_1$ (defined later), then we have

$$V_M^{\bar{\pi},T_\epsilon}(s) \geq V_{M_L}^{\bar{\pi},T_\epsilon}(s) - T_\epsilon R_{\max} \Pr(A_M) \tag{168}$$

$$\geq V_{M_L}^{\bar{\pi},T_\epsilon}(s) - T_\epsilon \epsilon_1 \tag{169}$$

$$\geq V_{\bar{M}_{\omega_X,L}}^{\bar{\pi},T_\epsilon}(s) - T_\epsilon \epsilon_1 - g(\eta_T, \eta_R) \tag{170}$$

$$\geq V_{\bar{M}_{Y,L}}^{\bar{\pi},T_\epsilon}(s) - T_\epsilon \epsilon_1 - g(\epsilon_2) - g(\eta_T, \eta_R) \tag{171}$$

$$\geq V_{\bar{M}_Y}^{*,T_\epsilon}(s) - T_\epsilon \epsilon_1 - g(\epsilon_2) - g(\eta_T, \eta_R) \tag{172}$$

$$\geq V_M^{*,T_\epsilon}(s) - T_\epsilon \epsilon_1 - g(\epsilon_2) - g(\eta_T, \eta_R) - 2g(\eta_T + \epsilon_2, \eta_R). \tag{173}$$

Here the step from equation 168 to equation 169 follows because of the assumption that $R_{\max} \Pr(A_M) < \epsilon_1$. The step from equation 169 to equation 170 follows from Lemma 6, where $g(\eta_T, \eta_R) = T_\epsilon \eta_R + \frac{(T_\epsilon - 1)T_\epsilon}{2} \eta_T |\bar{S}| R_{\max}$. The step from equation 170 to equation 171 follows from Lemma 8, where $g(\epsilon_2) = \frac{(T_\epsilon - 1)T_\epsilon}{2} \epsilon_2 |\bar{S}| R_{\max}$. The step from equation 171 to equation 172 follows because the R-MAX policy $\bar{\pi}$ is the optimal policy for $\bar{M}_{Y,L}$, and $\bar{M}_{Y,L}$ is the same as $\bar{M}_Y$ on the known state-action pairs and overestimates the value of the unknown state-action pairs (to the maximum value). Finally, the step from equation 172 to equation 173 follows from Lemma 11.

In equation 173 the results are for the undiscounted $T_\epsilon$-step sum of rewards, so to obtain the result for the average reward per step, we have to divide equation 173 by $T_\epsilon$. We get

$$\text{Opt}(\prod_M(\epsilon, T_\epsilon)) - T_\epsilon\epsilon_1/T_\epsilon - g(\epsilon_2)/T_\epsilon - g(\eta_T, \eta_R)/T_\epsilon - 2g(\eta_T + \epsilon_2, \eta_R)/T_\epsilon \tag{174}$$

$$= \text{Opt}(\prod_M(\epsilon, T_\epsilon)) - \epsilon_1 - \frac{(T_\epsilon - 1)T_\epsilon}{2}\epsilon_2|\bar{S}|R_{\max}/T_\epsilon$$

$$- (T_\epsilon\eta_R + \frac{(T_\epsilon - 1)T_\epsilon}{2}\eta_T|\bar{S}|R_{\max})/T_\epsilon - 2(T_\epsilon\eta_R + \frac{(T_\epsilon - 1)T_\epsilon}{2}(\eta_T + \epsilon_2)|\bar{S}|R_{\max})/T_\epsilon \tag{175}$$

$$= \text{Opt}(\prod_M(\epsilon, T_\epsilon)) - \epsilon_1 - \frac{(T_\epsilon - 1)}{2}\epsilon_2|\bar{S}|R_{\max}$$

$$- \eta_R - \frac{(T_\epsilon - 1)}{2}\eta_T|\bar{S}|R_{\max} - 2\eta_R - (T_\epsilon - 1)(\eta_T + \epsilon_2)|\bar{S}|R_{\max} \tag{176}$$

$$= \text{Opt}(\prod_M(\epsilon, T_\epsilon)) - \epsilon_1 - \frac{(T_\epsilon - 1)}{2}\epsilon_2|\bar{S}|R_{\max}$$

$$- 3\eta_R - \frac{(T_\epsilon - 1)}{2}\eta_T|\bar{S}|R_{\max} - (T_\epsilon - 1)\epsilon_2|\bar{S}|R_{\max} - (T_\epsilon - 1)\eta_T|\bar{S}|R_{\max} \tag{177}$$

$$= \text{Opt}(\prod_M(\epsilon, T_\epsilon)) - \epsilon_1 - 3\frac{(T_\epsilon - 1)}{2}\epsilon_2|\bar{S}|R_{\max} - 3\eta_R - 3\frac{(T_\epsilon - 1)}{2}\eta_T|\bar{S}|R_{\max} \tag{178}$$

$$= \text{Opt}(\prod_M(\epsilon, T_\epsilon)) - \epsilon_1 - 3\frac{(T_\epsilon - 1)}{2}\epsilon_2|\bar{S}|R_{\max} - 3\frac{g(\eta_T, \eta_R)}{T_\epsilon} \tag{179}$$

$$= \text{Opt}(\prod_M(\epsilon, T_\epsilon)) - \frac{3}{8}\epsilon - 3\frac{(T_\epsilon - 1)}{2}\epsilon_2|\bar{S}|R_{\max} - 3\frac{g(\eta_T, \eta_R)}{T_\epsilon} \tag{180}$$

$$= \text{Opt}(\prod_M(\epsilon, T_\epsilon)) - \frac{3}{8}\epsilon - 3\frac{(T_\epsilon - 1)}{2}\frac{3\epsilon}{4|\bar{S}|R_{\max}(T_\epsilon - 1)}|\bar{S}|R_{\max} - 3\frac{g(\eta_T, \eta_R)}{T_\epsilon}. \tag{181}$$

$$= \text{Opt}(\prod_M(\epsilon, T_\epsilon)) - \frac{3}{8}\epsilon - \frac{9}{8}\epsilon - 3\frac{g(\eta_T, \eta_R)}{T_\epsilon}. \tag{182}$$

$$= \text{Opt}(\prod_M(\epsilon, T_\epsilon)) - \frac{3}{2}\epsilon - 3\frac{g(\eta_T, \eta_R)}{T_\epsilon}. \tag{183}$$

In the step from equation 178 to equation 179 we use that $g(\eta_T, \eta_R) = T_\epsilon\eta_R + \frac{(T_\epsilon - 1)T_\epsilon}{2}\eta_T|\bar{S}|R_{\max}$. Then, in the last steps, we define $\epsilon_1$ and $\epsilon_2$. In the step from equation 179 to equation 180 we set $\epsilon_1 = \frac{3}{8}\epsilon$. And in the step from equation 180 to equation 181 we set $\epsilon_2 = 3\epsilon/(4|\bar{S}|R_{\max}(T_\epsilon - 1))$.

The above assumed that the algorithm did not fail, but we cannot guarantee this with probability 1 within a number of steps that is polynomial in the input. We will show that we can upper bound the probability of failure by $\delta$. There are three reasons why the algorithm could fail.

1. First, we need to show that the transition functions of $\bar{M}_Y$ are within $\eta_T + \epsilon_2$ of the transition functions of $\bar{M}_{\omega_X}$, with high probability. This is to ensure that, once all the abstract state-action pairs are known, the loss of value because of an inaccurate transition model, $V_{\bar{M}_Y}^{*,T_\epsilon} - V_M^{*,T_\epsilon}$ is within $2g(\eta_T + \epsilon_2, \eta_R) = 2T_\epsilon\eta_R + (T_\epsilon - 1)T_\epsilon(\eta_T + \epsilon_2)|\bar{S}|R_{\max}$ by Lemma 11. We can use the martingale concentration inequality to choose a number of samples $K_1$ so that the probability that our transition estimate is outside the desired bound is less than $\frac{\delta}{3|\bar{S}||A|}$ for every abstract state-action pair if we sample each pair $K_1$ times. By Lemma 10, we can guarantee this by using $K_1 \geq \frac{2[\ln(2^{|\bar{S}|} - 2) - \ln(\delta/(3|\bar{S}||A|))]}{(\frac{3\epsilon}{4|\bar{S}|R_{\max}(T_\epsilon - 1)})^2} = \frac{32|\bar{S}|^2 R_{\max}^2(T_\epsilon - 1)^2[\ln(2^{|\bar{S}|} - 2) - \ln(\delta/(3|\bar{S}||A|))]}{9\epsilon^2}$. Then, by applying the Union Bound on all $|\bar{S}|A$ pairs, we have that the total probability that any transition function is outside the desired bound is less than $\delta/3$.

2. Before we assumed that $R_{\max} \Pr(A_M) < \epsilon_1 (= \frac{3\epsilon}{8})$. Here we can show that after $K_2$ $T_\epsilon$-step trials where $R_{\max} \Pr(A_M) \geq \frac{3\epsilon}{8}$, all the abstract state-action pairs are visited at least $K_1$ times (become known) with a probability of at least $1 - \delta/3$ by using Hoeffding's Inequality. Let $X_i$ be the indicator variable that is 1 if we visit an unknown abstract state-action pair in a trial, and 0 otherwise. For the trials where

$$R_{\max} \Pr(X_i = 1) \geq \frac{3\epsilon}{8}$$
$$\Pr(X_i = 1) \geq \frac{3\epsilon}{8}/R_{\max},$$

we can use Hoeffding's Inequality to establish an upper bound, we have:

$$\Pr(\sum_{i=1}^{K_2}((\frac{3\epsilon}{8}/R_{\max}) - X_i) \geq K_2^{2/3}) = \tag{184}$$

$$\Pr(\frac{3\epsilon}{8}\frac{K_2}{R_{\max}} - \sum_{i=1}^{K_2} X_i \geq K_2^{2/3}) \leq e^{-\frac{2(K_2^{2/3})^2}{K_2}} \leq e^{-\frac{2(K_2^{2/3})^2}{K_2}} = e^{-2K_2^{1/3}}, \tag{185}$$

$$\Pr(\frac{3\epsilon}{8}\frac{K_2}{R_{\max}} - K_2^{2/3} \geq \sum_{i=1}^{K_2} X_i) \leq e^{-2K_2^{1/3}}. \tag{186}$$

After $K_2$ exploring episodes we want $\sum_{i=1}^{K_2} X_i$, the number of visits to unknown state-action pairs, to be $K_1|\bar{S}||A|$. So we can choose $K_2$ such that $\frac{3\epsilon}{8}\frac{K_2}{R_{\max}} - K_2^{2/3} \geq K_1|\bar{S}||A|$, and $e^{-2K_2^{1/3}} \leq \delta/3$ to guarantee that the probability of failing to explore enough is at most $\delta/3$.

3. Finally, the actual return may be lower than the expected return when we perform a $T_\epsilon$-step trial where we do not explore. We use Hoeffding's Inequality to determine the number of steps $K_3$ needed to ensure that the actual average return is within $\epsilon/2$ of $\mathrm{Opt}(\prod_M(\epsilon, T_\epsilon)) - \frac{3}{2}\epsilon - 3\frac{g(\eta_T, \eta_R)}{T_\epsilon}$. We need to choose $K_3$ so that the probability of obtaining an actual return that is smaller than the desired $\mathrm{Opt}(\prod_M(\epsilon, T_\epsilon)) - 2\epsilon - 3\frac{g(\eta_T, \eta_R)}{T_\epsilon}$ is at most $\delta/3$ within $K_3 = Z|\bar{S}|T_\epsilon$ exploitation steps, with some number $Z > 0$. Let $X_i$ denote the average return in the $i$-th exploitation step and $\mu$ the average expected return in an exploitation step so that $\mu$ is at least $\mathrm{Opt}(\prod_M(\epsilon, T_\epsilon)) - \frac{3}{2}\epsilon - 3\frac{g(\eta_T, \eta_R)}{T_\epsilon}$. Then

$$\Pr(\sum_{i=1}^{K_3}(\frac{\mu - X_i}{R_{\max}}) \geq K_3^{2/3}) \leq e^{-2\frac{(K_3^{2/3})^2}{K_3}} = e^{-2K_3^{1/3}}. \tag{187}$$

This means that, with a probability of at most $e^{-2K_3^{1/3}}$, the average return for $K_3$ exploitation steps is more than $\frac{R_{\max}}{K_3^{1/3}}$ lower than $\mu$:

$$\Pr(\sum_{i=1}^{K_3}(\frac{\mu - X_i}{R_{\max}}) \geq K_3^{2/3}) \leq e^{-2K_3^{1/3}}, \tag{188}$$

$$\Pr(K_3\frac{\mu}{R_{\max}} - \sum_{i=1}^{K_3}\frac{X_i}{R_{\max}} \geq K_3^{2/3}) \leq e^{-2K_3^{1/3}}, \tag{189}$$

$$\Pr(K_3\mu - \sum_{i=1}^{K_3} X_i \geq R_{\max}K_3^{2/3}) \leq e^{-2K_3^{1/3}}, \tag{190}$$

$$\Pr(\mu - \sum_{i=1}^{K_3}\frac{X_i}{K_3} \geq \frac{R_{\max}}{K_3^{1/3}}) \leq e^{-2K_3^{1/3}}. \tag{191}$$

We can now choose $Z$, so that $\epsilon/2 \leq \frac{R_{\max}}{(Z|\bar{S}|T_\epsilon)^{\frac{1}{3}}}$ and $e^{-2(Z|\bar{S}|T_\epsilon)^{1/3}} \leq \delta/3$, to get the desired result: with probability at most $\delta/3$ the obtained value will be more than $\epsilon/2$ lower than the expected value.

The probability of failure is thus at most $3 * \delta/3 = \delta$, and an average return at most $2\epsilon + 3\frac{g(\eta_T, \eta_R)}{T_\epsilon}$ lower than $\text{Opt}(\prod_M(\epsilon, T_\epsilon))$ will be obtained with a probability of at least $1 - \delta$. □

## E Simulator Data Collection

Here we assume that we have access to a simulator and use this in our procedure to give a guarantee in the form of the abstract L1 inequality from equation 21. To some extent, this is not surprising, but to the best of our knowledge, this is the first work that explicitly shows how to combine MBRL and abstraction using a simulator. We assume that the simulator allows us to select (or move to) any state and draw a sample from its transition function, which we call the independent samples assumption:

**Assumption 1** (Independent samples). *We assume we can obtain independent samples, e.g., for any state-action pair $(s, a)$, we can draw samples directly from its transition function $T(\cdot|s, a)$.*

| **Algorithm 2** Procedure: MBRLAO | **Algorithm 3** COLLECTSAMPLES with Simulator |
|---|---|
| **Input:** $M, \phi, \delta, \epsilon, \pi$ 
 $\bar{Y} = \text{COLLECTSAMPLES}(M, \phi, \delta, \epsilon, \pi)$ 
 The sampling results in sequences $\bar{Y}_{\bar{s}, a}$, one for every pair $(\bar{s}, a)$: 
 $\bar{Y}_{\bar{s}, a} = \phi(s'^{(1)}), \cdots, \phi(s'^{(m)})$ 
 $= \bar{s}'^{(1)}, \cdots, \bar{s}'^{(m)}$ 
 **for all** $(\bar{s}, a, \bar{s}') \in \bar{S} \times A \times \bar{S}$ **do** 
 $\quad \bar{T}_Y(\bar{s}'|\bar{s}, a) = \frac{1}{m} \sum_{i=1}^m \mathbb{1}\{\bar{Y}_{\bar{s}, a}^{(i)} = \bar{s}'\}$ 
 **end for** 
 $\bar{M}_Y \triangleq \langle \bar{S}, A, \bar{T}_Y, \bar{R}, \gamma \rangle$ 
 $\bar{\pi}_Y^* = \text{Value Iteration}(\bar{M}_Y)$ 
 Apply $\bar{\pi}_Y^*$ to $M$ | **Input:** $M, \phi, \delta, \epsilon$ 
 $\kappa = \frac{\delta}{|\bar{S}||A|}$ 
 $m = \lceil \frac{2[\ln(2^{|\bar{S}|} - 2) - \ln(\kappa)]}{\epsilon^2} \rceil$ 
 **for all** $(\bar{s}, a) \in \bar{S} \times A$ **do** 
 $\quad \bar{Y}_{\bar{s}, a} = [\ ]$ 
 $\quad x_{\bar{s}, a} = \text{select a prototype state } s \in \bar{s}$ 
 $\quad$ **for** $i = 1 : m$ **do** 
 $\quad \quad s' = \text{Sample}(T(\cdot|x_{\bar{s}, a}, a))$ 
 $\quad \quad \bar{Y}_{\bar{s}, a}.\text{append}(\phi(s'))$ 
 $\quad$ **end for** 
 **end for** 
 **Return:** all $\bar{Y}_{\bar{s}, a}$ |

If a simulator of the MDP is available, this is a reasonable assumption. For every pair $(\bar{s}, a)$, the simulator sampling procedure (Algorithm 3) selects a prototype $x_{\bar{s}, a} \in \bar{s}$ from which to sample. We define a weighting function $\omega_X(s, a)$ that has a weight of 1 if $s$ is the prototype $x_{\bar{s}, a}$ and 0 otherwise:

$$\forall_{(\bar{s}, a), s \in \bar{s}} \ \omega_X(s, a) \triangleq \mathbb{1}\{s = x_{\bar{s}, a}\}. \tag{192}$$

Then we use this $\omega_X$ to define the abstract transition function $\bar{T}_{\omega_X}$ according to equation 10. $\bar{T}_{\omega_X}(\bar{s}'|\bar{s}, a) = \sum_{s' \in \bar{s}'} T(s'|s = x_{\bar{s}, a}, a)$. This way, the samples we collect for one pair $(\bar{s}, a)$ are i.i.d. They are independent because of Assumption 1 and identically distributed because we sample from the prototype. Because the samples are i.i.d., we can use Lemma 1. We show that, with the simulator we can combine MBRL with abstraction and still learn an accurate model. We can guarantee that $\bar{T}_Y$ will be close to $\bar{T}_{\omega_X}$, with a high probability:

**Theorem 5.** *Under assumption 1, following the procedure in Algorithm 1, with the data collection from Algorithm 3 and inputs $|\bar{S}|, A, \epsilon,$ and $\delta$. For $\bar{T}_Y$ constructed by the algorithm, we have that with probability $1 - \delta$, the following holds:*

$$\forall_{(\bar{s}, a)} \ ||\bar{T}_Y(\cdot|\bar{s}, a) - \bar{T}_{\omega_X}(\cdot|\bar{s}, a)||_1 \leq \epsilon. \tag{193}$$

### E.1 Proof of Theorem 5

Before starting with the actual proof, we first go over Algorithm 3 and give two lemmas the proof uses.

The agent will draw samples using the simulator as described in Algorithm 3. Since we assume we can sample directly from the transition functions $T(\cdot|s, a)$, this algorithm loops over all pairs $(\bar{s}, a)$ and samples $m$ times[10]

---

[10] The value of $m$ in Algorithm 3 is chosen based on the results further along in this section.

from each transition function. More formally, for every pair $(\bar{s}, a)$, the algorithm selects one prototype state $x_{\bar{s},a} = s \in \bar{s}$. Then, it loops over every pair $(\bar{s}, a)$ and samples $m$ transitions from $T(\cdot | x_{\bar{s},a}, a)$. The set of collected experiences for each abstract state-action pair $(\bar{s}, a)$ is represented by $\bar{Y}_{\bar{s},a}$, as defined by equation 15.

Given $\bar{Y}_{\bar{s},a}$, we define the learned model $\bar{T}_Y(\cdot | \bar{s}, a)$ according to equation 16, $\bar{T}_{\omega_X}$ according to equation 19, and $\omega_X$ according to equation 192. It follows from Lemma 1 that we can derive a number of samples that we require to guarantee that $\Pr(||\bar{T}_Y(\cdot | \bar{s}, a) - \bar{T}_{\omega_X}(\cdot | \bar{s}, a)||_1 \geq \epsilon) \leq \kappa$ is true for inputs $\kappa$ and $\epsilon$:

**Lemma 4.** *For inputs $\kappa$ and $\epsilon$ ($0 < \kappa < 1, 0 < \epsilon < 2$), we have that the following holds for $m \geq \frac{2[\ln(2^{|\bar{S}|}-2)-\ln(\kappa)]}{\epsilon^2}$:*

$$\Pr(||\bar{T}_Y(\cdot | \bar{s}, a) - \bar{T}_{\omega_X}(\cdot | \bar{s}, a)||_1 \geq \epsilon) \leq \kappa. \tag{194}$$

*Proof.* To shorten the notation, we use the definitions $P_Y \triangleq \bar{T}_Y(\cdot | \bar{s}, a)$ and $P_{\omega_X} \triangleq \bar{T}_{\omega_X}(\cdot | \bar{s}, a)$. From Lemma 1, we have that

$$\Pr(||P_Y - P_{\omega_X}||_1 \geq \epsilon) \leq (2^{|\bar{S}|} - 2)e^{-\frac{1}{2}m\epsilon^2}. \tag{195}$$

We need to select $m$ such that $\kappa \geq (2^{|\bar{S}|} - 2)e^{-\frac{1}{2}m\epsilon^2}$:

$$\kappa \geq (2^{|\bar{S}|} - 2)e^{-\frac{1}{2}m\epsilon^2} \tag{196}$$

$$\frac{\kappa}{2^{|\bar{S}|} - 2} \geq e^{-\frac{1}{2}m\epsilon^2} \tag{197}$$

$$\ln(\kappa) - \ln(2^{|\bar{S}|} - 2) \geq -\frac{m\epsilon^2}{2} \tag{198}$$

$$\frac{m\epsilon^2}{2} \geq \ln(2^{|\bar{S}|} - 2) - \ln(\kappa) \tag{199}$$

$$m \geq \frac{2[\ln(2^{|\bar{S}|} - 2) - \ln(\kappa)]}{\epsilon^2} \tag{200}$$

Thus, if $m \geq \frac{2[\ln(2^{|\bar{S}|}-2)-\ln(\kappa)]}{\epsilon^2}$ we have

$$\Pr(||P_Y - P_{\omega_X}||_1 \geq \epsilon) \leq \kappa. \qquad \square$$

Using the Union bound, we can give a lower bound on the probability that $\bar{T}_Y(\cdot | \bar{s}, a)$ and $\bar{T}_{\omega_X}(\cdot | \bar{s}, a)$ are $\epsilon$ close for every $(\bar{s}, a)$:

**Lemma 5.** *If*

$$\forall_{(\bar{s},a)} \left[ \Pr(||\bar{T}_Y(\cdot | \bar{s}, a) - \bar{T}_{\omega_X}(\cdot | \bar{s}, a)||_1 \geq \epsilon) \right] \leq \frac{\delta}{|\bar{S}||A|} \tag{201}$$

*then the following holds with a probability of at least $1 - \delta$:*

$$\max_{(\bar{s},a)} \left[ ||\bar{T}_Y(\cdot | \bar{s}, a) - \bar{T}_{\omega_X}(\cdot | \bar{s}, a)||_1 \right] \leq \epsilon. \tag{202}$$

*Proof.* We define

$$\Delta_{\bar{s},a} \triangleq ||\bar{T}_Y(\cdot | \bar{s}, a) - \bar{T}_{\omega_X}(\cdot | \bar{s}, a)||_1. \tag{203}$$

Then $\Pr(\max_{(\bar{s},a)}\{\Delta_{\bar{s},a} \geq \epsilon\})$ is the probability that $\Delta_{\bar{s},a} \geq \epsilon$ for at least one abstract state-action pair. From the union bound, it follows that $\Pr(\max_{(\bar{s},a)}\{\Delta_{\bar{s},a} \geq \epsilon\}) \leq \delta$:

$$\Pr(\max_{(\bar{s},a)}\{\Delta_{\bar{s},a} \geq \epsilon\}) \leq \sum_{\bar{s},a} \Pr(\Delta_{\bar{s},a} \geq \epsilon) \tag{204}$$

$$\leq \sum_{\bar{s},a} \frac{\delta}{|\bar{S}||A|} \tag{205}$$

$$= \delta. \tag{206}$$

It follows that $\Pr(\max_{(\bar{s},a)}\{\Delta_{\bar{s},a} \leq \epsilon\}) \geq 1 - \delta$ since $\Pr(\max_{(\bar{s},a)}\{\Delta_{\bar{s},a} \leq \epsilon\}) = 1 - \Pr(\max_{(\bar{s},a)}\{\Delta_{\bar{s},a} \geq \epsilon\})$. Thus the probability that equation 202 holds is at least $1 - \delta$. $\qquad\square$

Now we are ready to proof Theorem 5:

*Proof of Theorem 5.* By Assumption 1, and the earlier assumption that $|S|$ and $|A|$ are finite, we have that we can obtain $m$ samples in finite time for every abstract state-action pair and any $m > 0$. Given the inputs $|\bar{S}|, A, \epsilon$, and $\delta$, Algorithm 3 sets $m = \lceil \frac{2[\ln(2^{|\bar{S}|}-2) - \ln(\kappa)]}{\epsilon^2} \rceil$, where $\kappa = \frac{\delta}{|\bar{S}||A|}$. Then, for every $(\bar{s}, a)$, a prototype state $x_{\bar{s},a} = s \in \bar{s}$ is selected. We use equation 192 to define $\omega_X$ and equation 19 to define $\bar{T}_{\omega_X}$.

For all $(\bar{s}, a)$, Algorithm 3 obtains a sequence $\bar{Y}_{\bar{s},a}$ by sampling from the transition function from the prototype state $x_{\bar{s},a}$ and Algorithm 2 constructs the empirical transition functions as in equation 16.

Given our choice of $m$ and the inputs $\kappa = \frac{\delta}{|\bar{S}||A|}$ and $\epsilon$, it follows from Lemma 10 that

$$\forall_{(\bar{s},a)} \ \Pr(||\bar{T}_Y(\cdot|\bar{s},a) - \bar{T}_{\omega_X}(\cdot|\bar{s},a)||_1 \geq \epsilon) \leq \frac{\delta}{|\bar{S}||A|}. \tag{207}$$

By the union bound, we have that the following holds with a probability of at least $1 - \delta$:

$$\forall_{(\bar{s},a)} \ ||\bar{T}_Y(\cdot|\bar{s},a) - \bar{T}_{\omega_X}(\cdot|\bar{s},a)||_1 \leq \epsilon. \tag{208}$$

$\qquad\square$

