# OpenReview forum: "An Analysis of Model-Based Reinforcement Learning From Abstracted Observations"
_TMLR — Accepted by TMLR_

### Review · Reviewer_6VFG · 2023-04-12

**Summary Of Contributions:**

The paper discusses model-based RL over abstracted states, specifically using a model-similarity/model-irrelevance state abstraction. The authors show that when the data is collected online, we cannot learn an accurate model of the abstract MDP because the samples from a given abstract state are not iid. They then show that samples being identically distributed is not required, and resolve for the indepence issue using martingale bounds. Finally, they use this insight to prove convergence for the abstracted version of the R-Max algorithm.

**Audience:**

No

**Claims And Evidence:**

Yes

**Requested Changes:**

N/A

**Strengths And Weaknesses:**

The paper is written-well and discusses prior work well. The problem is setup is fairly well motivated.

I think the paper deals with a very narrow setting, i.e. the negative result presented in the paper corresponds to a particular method category (model-based RL in this case) as well as a specific analysis. Furthermore, the main issue of concern is shown to be the sample independence when collecting data online, but that has little to do with the kind of state abstraction considered but more around the observation that the state abstaction (when not exact) leads to a POMDP setting. The actual issue with the model-irrelevance state abstraction seems to be the samples not being identical, but (as the authors show), that can be easily remedied. Therefore, overall I think the insights from the paper are too narrow and do not benefit a larger audience.

---

> ### Author Response · Authors · 2023-04-21
> **Response to Review of Paper796 by Reviewer 6VFG**
>
> We are glad to see that the reviewer agrees that in terms of technical correctness (the first criterion for TMLR) our paper is strong. Indeed, we explicitly rectify a common, incorrect belief that abstraction does not cause any issues for model-based RL, which has led to factual errors in the past.
>
> We would like to avoid getting into a debate on the definition of a "narrow setting", but would like to stress the generality of our results, to avoid any potential misconceptions:
>
> 1) Indeed, both the sample independence and the samples not being identical have little to do with the kind of state abstraction considered. It, therefore, is a general result: our negative results show a lack of independence, *for all forms of abstraction* (for instance, also including deep RL). This result is important for *all* reinforcement learning with abstraction.
>
> 2) The positive result shows that *for (approximate) model irrelevance abstractions*, things are fixable.
>
> Based on this, it is actually likely that there is a significant group of researchers that would be interested in this, and thus also the second criterion that it is of interest to “*some* individuals in TMLR's audience” is easily satisfied.

---

### Review · Reviewer_DDAA · 2023-04-24

**Summary Of Contributions:**

This work studies the setting of Abstracted Model-Based Reinforcement Learning. The authors introduce an analysis of the RMAX algorithm for this setting based on a martingale analysis.The authors claim the previous analyses relied on independence of samples and thus were not suitable for modification in the abstracted model based setting. The authors main contribution is to instead derive a martingale bound for abstracted MDPs that can be plugged in seamlessly into the analysis of algorithms such as RMAx to obtain PAC-like bounds for RL.

**Audience:**

Yes

**Broader Impact Concerns:**

No broader impact concerns.

**Claims And Evidence:**

No

**Requested Changes:**

The issues specified above are important. Responding to these would be paramount to decide wether this work should be admitted or not into TMLR.

**Strengths And Weaknesses:**

The main strength of this submission is that the paper is fairly well written. The prose is easy to follow.

The main weakness in my opinion is that it is really unclear how deriving a simple martingale bound is a novel or useful thing. Lots of existing analysis of online PAC and regret algorithms for Reinforcement Learning (see algorithms like UCBVI and others) already rely on martingale bounds for their analysis. This is in my opinion not really a surprising or new contribution. For example (Near-optimal Regret Bounds for Reinforcement Learning) already heavily uses martingale arguments in their analysis.

An extra concern is that it is unclear in definition 2 what is the nature of the $\omega$ function. This is a concern because then it is argued that $\omega_X$ will be an estimator of this quantity. It is unclear what this $\omega_X$ quantity will converge to because it depends on the sampling policies that will be used to explore the MDP. If at different points in time there are different ways of arriving to different (non-abstract) states, then the empirical estimators of $\omega_X$ will converge to different quantities. Because of this I find it very suspect that Theorem 4 does not have a dependence on $S$.

---

> ### Author Response · Authors · 2023-05-16
> **Response 1 to Review of Paper796 by Reviewer DDAA**
>
> *“The main weakness in my opinion is that it is really unclear how deriving a simple martingale bound is a novel or useful thing. Lots of existing analysis of online PAC and regret algorithms for Reinforcement Learning … already rely on martingale bounds for their analysis. This is in my opinion not really a surprising or new contribution.”*
>
> Indeed martingale concentration inequalities have been used more often in online RL analysis. Our novelty is in using it in the setting of RL plus (approximate) abstraction, where we use it in order to overcome some problems in this setting (dependence) and show that we can use it to provide performance guarantees in this setting. As far as we know, our paper is the first to do so.
>
> In itself, the observation that dependence crops up when abstracting is important, and is something that has not always been recognized or simply ignored [1,2, 3, 4, 5, 6, 7,8, 9,10, 11, 12, 13, 14], often by simply explicitly, or implicitly, assuming that the samples are independent. As our example shows samples cannot be assumed to be independent in this setting, analyses using this assumption do not hold.
> The martingale bound allows us to establish bounds in this specific setting. Specifically, we use martingale bounds in Theorem 2 to establish a concentration inequality for the (abstract) transition model, when the samples may be dependent.
> [2, 3, 4, 5, 6, 7, 9, 11, 12, 13, 14] do not use martingale concentration inequalities for the transition model, and instead use inequalities that assume independent samples, as such their results do not apply in the “Abstracted RL” setting.
> [1, 8, 10] do use martingales for a transition model, but they do not employ these to deal with the issue of dependent samples (and their analysis does not extend to that case). Instead, in [1] they are in a block-MDP setting, where the observation space itself is Markov.
> In [8] they do not deal with dependence since they state "... the random variables we consider for rewards and transition probabilities are independent". [10] follows the steps of [8], and as we write, in their setting the MDP is fixed given the timestep, but in the abstraction setting this is not fixed, each time we run the MDP the transition function at a timestep *t* could be different.
>
> [1] Kamyar Azizzadenesheli, Alessandro Lazaric, and Animashree Anandkumar. Reinforcement learning in rich-observation mdps using spectral methods. arXiv preprint, 2016.
>
> [2] Hu, Y., Ji, Z., & Telgarsky, M. (2021). Actor-critic is implicitly biased towards high entropy optimal policies. arXiv preprint.
>
> [3] Thomas Jaksch, Ronald Ortner, and Peter Auer. Near-optimal regret bounds for reinforcement learning. Journal of Machine Learning Research, 11(Apr):1563–1600, 2010.
>
> [4] Tor Lattimore, Marcus Hutter, Peter Sunehag, et al. The sample-complexity of general reinforcement learning. In Proceedings of the 30th International Conference on Machine Learning. Journal of Machine Learning Research, 2013.
>
> [5] Lihong Li. A unifying framework for computational reinforcement learning theory. PhD thesis, Rutgers University-Graduate School-New Brunswick, 2009.
>
> [6] Odalric-Ambrym Maillard, Phuong Nguyen, Ronald Ortner, and Daniil Ryabko. Optimal
> regret bounds for selecting the state representation in reinforcement learning. In International Conference on Machine Learning, pages 543–551. PMLR, 2013.
>
> [7] Ronald Ortner, Odalric-Ambrym Maillard, and Daniil Ryabko. Selecting near-optimal approximate state representations in reinforcement learning. In International Conference on
> Algorithmic Learning Theory, pages 140–154. Springer, 2014.
>
> [8] Ronald Ortner, Daniil Ryabko, Peter Auer, and Rémi Munos. Regret bounds for restless markov bandits. Theoretical Computer Science, 558:62–76, 2014.
>
> [9] Ronald Ortner, Matteo Pirotta, Alessandro Lazaric, Ronan Fruit, and Odalric-Ambrym Maillard. Regret bounds for learning state representations in reinforcement learning. In Advances in Neural Information Processing Systems, pages 12738–12748, 2019.
>
> [10] Ronald Ortner, Pratik Gajane, and Peter Auer. Variational regret bounds for reinforcement learning. In Uncertainty in Artificial Intelligence, pages 81–90. PMLR, 2020.
>
> [11] Song, Z., Mei, S., & Bai, Y. (2021). When Can We Learn General-Sum Markov Games with a Large Number of Players Sample-Efficiently? arXiv preprint.
>
> [12] Saha, A., & Krishnamurthy, A. (2022, March). Efficient and Optimal Algorithms for Contextual Dueling Bandits under Realizability. In International Conference on Algorithmic Learning Theory (pp. 968-994). PMLR.
>
> [13] Alexander L Strehl and Michael L Littman. An analysis of model-based interval estimation for markov decision processes. Journal of Computer and System Sciences, 74(8), 2008.
>
> [14] Zhang, X., Song, Y., Uehara, M., Wang, M., Agarwal, A., & Sun, W. (2022, June). Efficient reinforcement learning in block mdps: A model-free representation learning approach. In International Conference on Machine Learning. PMLR.

---

> ### Author Response · Authors · 2023-05-16
> **Response 2 to Review of Paper796 by Reviewer DDAA**
>
> *“An extra concern is that it is unclear in definition 2 what is the nature of the $\omega$ function. This is a concern because then it is argued that $\omega_x$  will be an estimator of this quantity. It is unclear what this $\omega_x$ quantity will converge to because it depends on the sampling policies that will be used to explore the MDP. If at different points in time there are different ways of arriving to different (non-abstract) states, then the empirical estimators of $\omega_x$ will converge to different quantities. Because of this I find it very suspect that Theorem 4 does not have a dependence on $S$.”*
>
> $\omega_x$ is used in the bounds where we estimate the transition towards abstract states, e.g., Theorem 2, which establishes a concentration inequality for the abstract transition function. In there, even though $\omega_x$ will change over time and indeed depends on the sampling policies. The beauty of our analysis is that it does not matter which $\omega_x$ it converges to, or if it even converges. Instead, this Theorem shows that the difference between the empirical abstract transition function and an abstract transition function based on some (particularly chosen)  $\omega_x$ will converge to (almost) 0. Here $\omega_x$ is constructed from the state-action pairs that are encountered. The dependence here is on $\bar{S}$ instead of $S$ because the experiences from the different states are grouped together.
> Theorem 4 uses the result of Theorem 2 to establish that, with probability, an accurate abstract transition model is learned. With this, the performance bounds from Theorem 1 can be used, since these hold for *any* $\omega_x$, as long as $\omega_x$ is a valid weighting function.
> $\omega$ (and $\omega_x$) is valid as long as the weights of all $(s, a)$ are between 0 and 1 ($0 <= \omega(s, a) <= 1$), and the weights of an abstract state-action pair sum up to 1 ($\forall \bar{s} \sum_{s’ \in \bar{s}} \omega(s’, a) = 1$).
> That $\omega_x$ will be a valid weighting function follows from its definition.
>
> Though we respect the reviewer's feeling of suspicion, in order for us to potentially do something about it, the reviewer needs to point out where exactly the proof goes wrong. We checked the proof again and, at this point, we remain convinced the proof and the statement in Theorem 4 are correct.

---

> > ### Comment · Reviewer_DDAA · 2023-05-30
> > **Still unconvinced**
> >
> > Dear authors,
> >
> > I really appreciate your response. I don't think the $w_x$ argument is correct. These quantities as defined in the text refer to empirical quantities that have a changing distribution depending on what step is the algorithm's run at. In many online algorithms the sampling (policy) distribution over states changes in every step. In fact the RMAX algorithm also does this, something that is completely obfuscated in Algorithm 1 with the notation Step(s,a) [this should have been time-indexed]. This makes it impossible for these $\omega_x$ weights to converge to any meaningful population quantity. It is thus also unclear what $T_Y$ may be as the population weights are never defined. For $T_Y$ to make any sense this has to be defined with respect to a set of a-priori weights.
> >
> > As for the main claim f the paper that using martingales is a novel idea I am still unconvinced. Most of the modern analysis of theoretical MBRL heavily rely on martingale inequalities precisely because the samples are not i.i.d. (they are Markov). In a simple RL trajectory of the form $s, a, s', a', s, a, s''$ the states $s''$ and $s'$ are not i.i.d, all we can say is their conditional distributions $s'' \sim P(\cdot |s,a)$ and $s' \sim P(\cdot | s,a)$ are the same. This is already something that is a challenge in the 'usual' RL setting. I say this because the premise of writing Lemma 1 is moot. As it stands I cannot recommend acceptance of this work to TMLR. These clarity issues need to be addressed before, particularly the nature of $T_Y$, and the weights $\omega_x$. I do't think the argument makes any sense without these $\omega_x$ being defined in advance explicitly. I also don't think it is correct to define them empirically because any algorithm would have a drifting distribution for the sampling policy. The authors response on the matter is lacking:
> >
> > " The beauty of our analysis is that it does not matter which  it converges to, or if it even converges. Instead, this Theorem shows that the difference between the empirical abstract transition function and an abstract transition function based on some (particularly chosen)  will converge to (almost) 0."
> >
> > This makes no sense. If $\omega_x$ was defined in advance completely adversarially, your results would still hold that the sampling procedure, that is completely independent of this $\omega_x$ would lead to an empirical abstract model that converges to $T_Y$ defined by $\omega_x$. This would imply that all $T_Y$ is equal for all $\omega_x$ choices something that is clearly not true.
> >
> > Because of these fundamental flaws I cannot recommend acceptance. I encourage the authors to revise the manuscript and either clarify these issues and resubmit to either this venue or another one.

---

> > > ### Comment · Reviewer_gAF9 · 2023-05-31
> > > **Thoughts on the $\omega_X$ argument**
> > >
> > > Hi all,
> > >
> > > Thanks for the discussion. I really appreciate Reviewer DDAA's careful read here. I think Reviewer DDAA makes a compelling point regarding the use of $\omega_X$. By my reading, Theorem 2 is intended to hold for a fixed but arbitrary choice of $N(\bar{s},a)$, which in turn defines $\omega_{X}, T_{Y}$, and $T_{\omega_X}$, since $\omega_X = \frac{1}{N(\bar{s},a)} ...$, and $T_Y = \frac{1}{N(\bar{s},a)} ...$. As I understand it, this ensures that the underlying weighting function for the two transition functions in Theorem 2 is identical, which is used in the move from Equation 116 to Equation 117 in the proof. In this sense, if an adversary set the weighting function $\omega_X$, it would also change $T_Y$. Does that seem right?
> > >
> > > However, it does appear that this is might raise an issue---the point Reviewer DDAA identified: "For $T_Y$ to make any sense this has to be defined with respect to a set of a-priori weights", whereas at the moment $\omega_X(\bar{s},a)$ is defined using (what I understand to be) the actual counts of the number of times each ground state $s \in \bar{s}$ was visited when action $a$ was executed in $\bar{s}$. Is this problematic? Any thoughts?
> > >
> > > Thanks,
> > >
> > > Reviewer gAF9

---

> > > > ### Author Response · Authors · 2023-06-04
> > > > **Response to the technical questions**
> > > >
> > > > We also appreciate the reviewers’ efforts in focusing on technical rigor. This is precisely what we had hoped for in submitting to TMLR, and so we are happy to see these detailed questions.
> > > >
> > > > Reviewer gAF931, you are right in your interpretation. We will slightly more explicitly summarize, to also address some of the questions by Reviewer DDAA30:
> > > >
> > > > - $\omega$ is defined first on p4. Please note that Theorem 1 describes the results for an abstract MDP $\bar{M}_\omega$ for *any* $\omega$.
> > > > - As such, below (13), we make the argument that IF we can show that $\bar{T}_{Y}$ is close to $\bar{T}_\omega$, for any valid $\omega$, we can give guarantees.
> > > > - Indeed, $\omega_X$, as defined above (15) is an empirical quantity that changes at every time step. However, at every time step, it *is* a valid $\omega$.
> > > > - Similarly $\bar{T}\_{Y}$ and $ \bar{T}_{ \omega_X }$ are quantities that change every time step. None of this matters, as long as we can have a concentration bound on their L1 distance, as formulated in (16). We can then make an appeal directly that with high probability the performance loss is bounded by appealing on Theorem 1.
> > > > - Our Theorem 2 shows that in fact (16) holds. Its proof is detailed in Appendix C.
> > > >
> > > > Returning to Reviewer gAF931’ question “Is this problematic? Any thoughts?”, we do not understand this (“has to be defined with respect to a set of a-priori weights”) comment. $\bar{T}_{Y}$ is simply defined by (13).
> > > >
> > > > Our result just shows that whatever $\bar{T}\_{Y}$ you might end up with (indeed, regardless of changing policies, etc.), it was generated by some underlying states $X$, and the implied $\bar{T}_{\omega_X}$ will concentrate on $\bar{T}\_{Y}$. We are of course happy to further clarify this in a final version.
> > > >
> > > > We hope this answers the questions about the correctness of our result. If any further doubts remain, we are of course happy to discuss them, but it would be helpful to then try and point to specific points in our proof that seem problematic.
> > > >
> > > > With respect to i.i.d. vs Markov, reviewer DDAA30 is right: without additional assumptions, the states are not i.d.d. We tried to make our main point, without getting lost in details (which we give below), but see that this has led to confusion, and we will clarify this in a final version.

---

> > > > > ### Author Response · Authors · 2023-06-04
> > > > > **Details on i.i.d. vs Markov**
> > > > >
> > > > > Let us explore the transitions from a particular state, say state 42, in a Markov chain (we can ignore actions for this argument). Let $k$ and $l$ denote the time steps of two different visits to state 42. We already agree that the conditional distributions from which next states are sampled are identical, so the question now is if these are independent. That is, is it the case that:
> > > > >   $P( S_{k+1}, S_{l+1} | S_k=42, S_l=42 ) = P( S_{k+1} | S_k=42 ) * P( S_{l+1} | S_l=42 )$?
> > > > >
> > > > > We have that
> > > > > $P( S_{k+1}, S_{l+1} | S_k=42, S_l=42 ) = $
> > > > > $P( S_{k+1} | S_k=42, S_l=42 ) P( S_{l+1} | S_k=42, S_k+1, S_l=42 ) =$
> > > > > $P( S_{k+1} | S_k=42, S_l=42 ) P( S_{l+1} | S_l=42 )$
> > > > > (due to the Markov property)
> > > > >
> > > > > So the question is if $P( S_{k+1} | S_k=42, S_l=42 )=P( S_{k+1} | S_k=42 )$?
> > > > > In general, this is not the case, since the information that $S_l=42$ gives information about what $S_{k+1}$ was.
> > > > >
> > > > > However, as shown for instance by Strehl & Littman (2008), concentration inequalities for i.i.d. samples, such as Hoeffding’s Inequality, can still be used as an upper bound in this case, indeed because of the Markov property.
> > > > >
> > > > > That result, however, does not apply to our setting. Specifically, it requires each sample to be identically distributed. Since we group the outcomes of multiple states together, the situation is more complicated. Without abstraction, we only need to consider $(s,a)$ and the next states $s’ \sim P(\cdot | s, a)$, which indeed have the same distribution. With abstraction, for $(\bar{s},a)$, we need to consider both the state $s \in \bar{s}$ that we reach and the resulting next state $\bar{s}’$. So with abstraction we consider $s’ \sim \sum\_{s \in \bar{s}} P(s,a| \pi, \text{history}) P(\cdot | s,a)$.
> > > > >
> > > > > The outcome of $\sum\_{s \in \bar{s}} P(s, a | \pi, \text{history})$ directly determines $\omega\_X$ and is dependent on the history and the policy $\pi$, the outcome of $P(\cdot | s,a)$ determines $\bar{T}\_Y$.
> > > > >
> > > > > Additionally, concentration inequalities for i.i.d. samples (such as the result from Weissman et al. (2003) and others) are often directly applied to the empirical transition function, e.g., Brafman & Tennenholtz (2002), Jaksch et al. (2010), Fruit et al. (2018), Bourel et al. (2020), without mentioning that these samples in a simple RL trajectory may not be independent. In our related work section we have discussed the use of concentration inequalities for i.i.d. samples in RL with abstracted observations and also discussed the result from Strehl & Littman (2008), we will discuss this more explicitly for RL in general.

---

> > > > ### Comment · Reviewer_DDAA · 2023-06-06
> > > > **Thanks for the engagement!**
> > > >
> > > > This is my main question.

---

> > > > > ### Author Response · Authors · 2023-06-07
> > > > > **Clarification**
> > > > >
> > > > > *"This is my main question."*
> > > > >
> > > > > It is not possible for us to see what specifically you react to, and thus we cannot understand what your main question is, and whether it still needs an answer from our side.
> > > > >
> > > > > We are happy to elaborate if you can clarify.

---

> > > > > > ### Comment · Reviewer_DDAA · 2023-06-19
> > > > > > **Follow up**
> > > > > >
> > > > > > Apologies for what the authors saw as a cryptic message. I was referring to the points raised by reviewer gAF9. I want to thank the authors for their comments. I see the point the authors are making about the $w_X$ weights. I still think this is problematic as the final result is of the form "our results hold for an abstract MDP with weights defined by the algorithm" as opposed to the ground truth. The concept of ground truth (algorithm independent $w_X$ weights that define the wold beyond the behavior of the given algorithm is completely abandoned. This would be akin to writing results for traditional MDP settings where the convergence is not measured with respect to the true MDP dynamics but w.r.t. how the algorithm behaves, something that changes from one algorithm to another or even between the execution of a given algorithm. Because of this I remain unconvinced.

---

> > > > > > > ### Author Response · Authors · 2023-06-20
> > > > > > > **Thank you for the follow up**
> > > > > > >
> > > > > > > We appreciate your willingness to engage in discussion. However, it seems that we are now at the point where we are no longer discussing our paper and the technical correctness of its claims. Indeed, our analysis is based on the insight that, whatever the sequences of underlying states would be (due to the stochastics of the environment and possibly the learning algorithm), it would correspond to some $\omega_X$ abstraction. This gives us the claimed results, and it seems that we now agree on this.
> > > > > > >
> > > > > > > Whether or not future methods should try to estimate the exact ground truth is an interesting philosophical question that our paper does not address. It is not clear what a ground truth set of weights should be here, or if it even exists, since the problem corresponds to a POMDP (with the underlying MDP and the abstraction as the observation function) and not to an abstract MDP with a fixed set of weights. Our paper only states that learning with an approximate model similarity abstraction will lead to only bounded performance loss, and in that sense is a good model of "approximate ground truth".
> > > > > > >
> > > > > > > If our interpretations are shared, we would appreciate it if you can acknowledge. If there are any remaining questions from your side that we did not address, please let us know.

---

> > > > > > > > ### Comment · Reviewer_DDAA · 2023-06-26
> > > > > > > > **Follow up**
> > > > > > > >
> > > > > > > > Dear authors,
> > > > > > > >
> > > > > > > > Thanks for your comments. I broadly agree with your first paragraph. Regarding the second. I don't think this is a merely philosophical question that can be simply brushed under the rug. At least an explicit detailed and not dismissive discussion about these alternatives, why are they hard (why not learning the ground truth if there is one?) is in order. Otherwise I don't see why is useful to do abstract RL when instead we can just use the original MBRL algorithm without the empirically derived abstractions. What is the performance difference of doing so? What is the edge in deriving an arbitrary data based / algorithm based abstraction if the bound will be worse than not using these clusterings. If there was a ground truth these questions would be easily answered, as it stands I am not sure what the answer is.'
> > > > > > > >
> > > > > > > > Thanks!

---

> > > > > > > > > ### Author Response · Authors · 2023-06-27
> > > > > > > > > **Follow up Response**
> > > > > > > > >
> > > > > > > > > From your reaction, we understand that with "estimating ground truth", you might mean "estimating the MDP without any form of abstraction". (We stand by our comment that with abstraction, there may not be a ground truth set of weights, since this is a POMDP). Certainly, without any abstraction, the question of weights etc. all becomes irrelevant, and all normal theory of model-based RL just applies.
> > > > > > > > >
> > > > > > > > > Indeed, even though we in this article take using abstraction for granted, this does mean that there is a tradeoff: a coarser abstracted model can potentially learn much faster but could sacrifice optimality, while a non-abstracted MDP might have the best performance in the limit of infinite experience. We explore this in section 3.3, where we compare the performance of R-MAX with and without abstraction. Indeed the performance bound with abstraction is worse when the abstraction is not exact, but it is a trade-off with the required amount of data (the number of steps is polynomial only in the number of abstract states).
> > > > > > > > >
> > > > > > > > > We are happy to clarify these issues further in a final draft.

---

### Review · Reviewer_gAF9 · 2023-05-04

**Summary Of Contributions:**

This paper studies state abstraction in model-based reinforcement learning (MBRL). The primary focus of the work is on combining the dimensionality reduction offered by state abstraction with model-based planning into a learning algorithm (typically those that rely on  exploration strategies that center around planning, such as R-Max). However, the work highlights the fact that in MBRL with state abstraction, i.i.d. sampling of the appropriate data cannot be assumed, contrary to the assumptions required to make use of standard concentration inequalities like Heoffding's and Wiessman et al.'s. In light of this, the paper draws on more general concentration inequalities for martingales that yield the same kinds of simulation-lemma like results we would expect in tabular MBRL. The first result, in my opinion, are observations 1, 2, and the counterexample. While similar issues have been identified in the past, I believe it is quite useful to have a clear counterexample such as this. Then, Section 3.2 explores abstracted MBRL beyond the i.i.d. assumption: martingales are introduced alongside Azuma's inequality, which is then used for a concentration inequality on model accuracy. These ideas are combined to produce Abstracted R-Max (Algorithm 1), along with its analysis, which together constitute the main results of the work (Theorems 3 and 4).

**Audience:**

Yes

**Broader Impact Concerns:**

None.

**Claims And Evidence:**

Yes

**Requested Changes:**

My only high-level comment and suggested change is to carefully rethink the term "Abstracted RL". I am not sure that the name "Abstracted RL" is buying you that much at the moment. Personally, I find it slightly confusing, as it is unclear whether it is describing a _problem_, or a class of _solution_ to the RL problem. It is introduced as a "setting", but I find this slightly vague. I do find the figure helpful (Figure 1). I suggest either dropping the term, or defining the term more precisely. For instance, when you say " In fact, none of the existing analyses of MBRL apply to Abstracted RL", is this suggesting that guarantees for existing algorithms do not hold under a specific set of conditions? If so, what exactly are these conditions? I could charitably interpret Figure 1 to include the RL problem when $\phi$ is the identity function, for instance. Is it specifically the case when $|\bar{S}| << |S|$? This still wont quite fully specify the problem as distinct from the RL problem when the underlying MDP is uncompressed.

Second, I include a small number low-level writing suggestions below that I believe will strengthen the paper:

_1. Introduction_:
- "((e.g., ...)": You can remove the double parenthetical and the "e.g." here, and just state "...typically (Strehl & Littman...".
- "The outline is as follows:...": Since it is only a 12 page paper, I am not sure the outline is strictly necessary. If you need space, I would encourage removing this paragraph, as well as the first paragraph of the Background section.

_2. Background_:
- You might introduce $R_{max}$ explicitly alongside the MDP and reward function since it will appear later.

**Strengths And Weaknesses:**

I was a reviewer on a previous version of this paper, and maintain a similarly positive outlook on the work. Additionally, many of the low-level suggestions from my earlier review have been addressed. As such, I still believe in what I had earlier identified as strengths. At the same time, some of the weaknesses of the work have since been addressed, and I believe the present work is a strong contribution that some among the TMLR community will find valuable.

[STRENGTHS]

Overall, this paper is clean, focused, and delivers a new result combining two prominent ideas in RL. While variations of some of the pieces of these ideas have appeared in the literature, I believe what is offered in this paper is novel and could be of interest to the community. Moreover, the survey of literature is sufficiently both broad and deep. Most relevant work is cited and discussed.

Clarity: First, a note on clarity. The work takes care in developing its main ideas, and is thorough in laying out the important definitions. Each definition is precisely stated with clean notation. I found the whole paper easy to follow.

Novelty: To my knowledge, the main results (Theorem 3/4) are novel, and I believe are nice results for the community in tabular MBRL and state abstraction to be aware of. Some of the underlying ideas (moving from i.i.d. to martingales) have been explored before in similar analysis, and naturally a big focus of MBRL is in "planning in latent space" (Hafner et al. 2019), but this particular set of insights and results are new.

The primary TMLR criteria are: (1) "Are the claims made in the submission supported by accurate, convincing and clear evidence?", and (2) "Would some individuals in TMLR's audience be interested in the findings of this paper?". I find the answer to both questions, "yes", and consequently, I recommend acceptance.


[WEAKNESSES]

The scope of the results is potentially limited. Aggregation functions capture only a small subset of possible ways to model state representation, and as such, it is not immediately clear what kind of applied algorithmic insight we might gain from the work. In light of this, I still find the main ideas and analysis useful and interesting. I do appreciate Section 5 that hints at possible future directions, but I still believe further expanding this section to present a broader consideration of how the work might yield new insights could be useful. For instance, how might the shift from the i.i.d. assumption to the martingale assumption impact algorithm designers that are interested in developing deep MBRL approaches? A few words in this direction could be useful. Are there any practical considerations, guidelines, or new predictions we can draw? This is not strictly necessary for publication, but I believe could strengthen the work.

---

> ### Author Response · Authors · 2023-05-17
> **Response to Review of Paper796 by Reviewer gAF9**
>
> Thank you for the in-depth review and suggestions, we will include the suggested minor changes.
>
> *“I do appreciate Section 5 that hints at possible future directions, but I still believe further expanding this section to present a broader consideration of how the work might yield new insights could be useful.”*
>
> We thank the reviewer for this suggestion and offer to rewrite the end of the discussion in the following way:
>
> Theorem 4 shows that, despite problems with dependence, we can give finite-sample guarantees when combining approximate model similarity abstractions with model-based RL. For good abstraction functions, i.e., when $\eta_R$ and $\eta_T$ are small and $|S| ≫ |\bar{S}|$, this leads to near-optimal solutions while needing fewer samples, compared to learning without abstraction. Practically, for tabular methods, these results mostly mean that concentration inequalities for independent samples have to be replaced in RL with abstract observations, for example by concentration inequalities based on martingales, as we have shown here.  In deep model-based RL, several recent empirical works have shown promising results by focusing on learning exact abstractions (van der Pol et al., 2020; Biza & Platt, 2019). An interesting direction is adapting these methods to learn approximate abstractions instead of exact abstractions. Since, compared to exact model similarity abstractions, approximate model similarity abstraction generally results in a smaller (abstract) state space; this could lead to faster learning.
>
> Our results shed further light on the observation from Abel et al. (2018) that RL with abstract observations is different from performing RL in an MDP constructed with abstraction. As our observations show, in RL with abstract observations the transition functions are not static, the samples are not identically distributed, and cannot be guaranteed to be independent. This could mean that in situations where we want to learn an abstraction, the behavior is also not quite as expected. In such situations, similar approaches that we applied here may prove useful, as many situations in RL have already been shown to not be independent processes. While our results hold for approximate model similarity models, there could be even more compact representations for which our techniques could lead to similar results. One clear example would be abstractions that focus not on state abstraction, but rather on state-action abstraction, of which state abstraction is simply a special case.
>
> People have been applying MDPs and RL to all kinds of problems, even though we know that the Markov property very rarely holds. Given that almost all theory of RL critically depends on this property, one could wonder why these things even work? Intuitively, we expect that the states in these successful applications are somehow "Markovian enough".
> In this work, we provide an understanding of this vague concept. Specifically, we show that an existing criterion of state representations ("approximate model similarity") in fact is a formal notion of "Markovian enough" in MBRL. Thus it provides critical insight into under what circumstances (and therefore in what applications) MBRL methods are expected to work.
>
>
> *“My only high-level comment and suggested change is to carefully rethink the term "Abstracted RL". I am not sure that the name "Abstracted RL" is buying you that much at the moment. Personally, I find it slightly confusing, as it is unclear whether it is describing a problem, or a class of solution to the RL problem.”*
>
> We agree and suggest the term “RL from abstracted observations” instead. In regular RL, the agent takes action and then observes a state s. With abstract observations, the agent instead observes an abstract state $\bar{s} (= \phi(s))$,  where the abstraction is the result of an (unknown) abstraction function $\phi$. $\phi$ being the identity function (or an exact model-similarity function) is a special case of this.

---

### Author Response · Authors · 2023-09-28
**Uploaded camera ready version**

Dear Reviewers and Editorial Team,

Thank you for the careful discussion and time spent reviewing our work. We are grateful for the opportunity to publish our work at TMLR and have uploaded a camera ready version.

Kind regards,
the authors.

---

### Decision · Action_Editors · 2023-09-05

**Recommendation:** Accept with minor revision

**Comment:**

The primary issue raised in the discussion repeatedly about the main theorem of the paper was that it proves convergence of a certain estimator to a quantity defined by the algorithm presented in the paper and the reviewer was not convinced that this result in particular would be of significant use in the community. I agree with this review and the full significance of this result is not clear. The authors have not been able to convince the reviewer about this point despite a revision.

My recommendation for this paper is based on the TMLR evaluation criteria and further discussion with the TMLR chief editors -

1. Are the claims made in the submission supported by accurate, convincing and clear evidence?
2. Would at least some individuals in TMLR's audience be interested in knowing the findings of this paper?

Criteria 1 is clear from the reviews and discussion. All the reviewers understand the main result in the paper and agree on its correctness. Criteria 2 remains unclear. The main question is whether the main result of the paper is itself of value and has the paper done enough to bring that out. The topic of the paper is clearly relevant to the community. Upon consideration of all the reviews, discussion and revision and upon conferring with the chief editors, I believe that while the paper is not completely upto the mark on point 2 of the criteria, there will be some members of the community who might be interested in the result (eg one of the reviewers). Therefore I am recommending a borderline Accept.

I suggest the authors take into account the suggestions made by the reviewer regarding the significance of the result and understanding how to further elaborate and strengthen the usefulness of their result.

**Audience:**

I believe that there is a fraction of the RL community that will be interested in the findings of this work. Please further read my comments below.

**Claims And Evidence:**

The reviewers discussed this paper in detail and came to the conclusion that the results AS STATED in the paper are correct and are supported by accurate convincing and clear evidence.

**Resubmission Of Major Revision:**

The authors may consider submitting a major revision at a later time.